# Omega-3 PUFAs prevent bone impairment and bone marrow adiposity in mouse model of obesity

Andrea Benova [1,2], Michaela Ferencakova[1], Kristina Bardova[3], Jiri Funda[3], Jan Prochazka [4], Frantisek Spoutil[4], Tomas Cajka [5], Martina Dzubanova[1,2], Tim Balcaen[6,7,8], Greet Kerckhofs [6,7,9,10], Wouter Willekens[11], G. Harry van Lenthe [12], Arzuv Charyyeva [1], Glenda Alquicer[1], Alena Pecinova[13], Tomas Mracek [13], Olga Horakova[3], Roman Coupeau [1], Morten Svarer Hansen[14], Martin Rossmeisl[3], Jan Kopecky [3] & Michaela Tencerova [1✉]

Obesity adversely affects bone and fat metabolism in mice and humans. Omega-3 poly-unsaturated fatty acids (omega-3 PUFAs) have been shown to improve glucose metabolism and bone homeostasis in obesity. However, the impact of omega-3 PUFAs on bone marrow adipose tissue (BMAT) and bone marrow stromal cell (BMSC) metabolism has not been intensively studied yet. In the present study we demonstrated that omega-3 PUFA supplementation in high fat diet (HFD + F) improved bone parameters, mechanical properties along with decreased BMAT in obese mice when compared to the HFD group. Primary BMSCs isolated from HFD + F mice showed decreased adipocyte and higher osteoblast differentiation with lower senescent phenotype along with decreased osteoclast formation suggesting improved bone marrow microenvironment promoting bone formation in mice. Thus, our study highlights the beneficial effects of omega-3 PUFA-enriched diet on bone and cellular metabolism and its potential use in the treatment of metabolic bone diseases.

[1] Laboratory of Molecular Physiology of Bone, Institute of Physiology of the Czech Academy of Sciences, Prague, Czech Republic. [2] Faculty of Science, Charles University, Prague, Czech Republic. [3] Laboratory of Adipose Tissue Biology, Institute of Physiology of the Czech Academy of Sciences, Prague, Czech Republic. [4] Czech Centre for Phenogenomics & Laboratory of Transgenic Models of Diseases, Institute of Molecular Genetics of the Czech Academy of Sciences, Prague, Czech Republic. [5] Laboratory of Translational Metabolism, Institute of Physiology of the Czech Academy of Sciences, Prague, Czech Republic. [6] Biomechanics lab, Institute of Mechanics, Materials, and Civil Engineering, UCLouvain, Louvain-la-Neuve, Belgium. [7] Pole of Morphology, Institute for Experimental and Clinical Research, UCLouvain, Brussels, Belgium. [8] Department of Chemistry, Molecular Design and Synthesis, KU Leuven, Leuven, Belgium. [9] Department of Materials Engineering, KU Leuven, Leuven, Belgium. [10] Prometheus, Division of Skeletal Tissue Engineering, Katholieke Universiteit Leuven, Leuven, Belgium. [11] FIBEr, KU Leuven, Leuven, Belgium. [12] Department of Mechanical Engineering, KU Leuven, Leuven, Belgium. [13] Laboratory of Bioenergetics, Institute of Physiology of the Czech Academy of Sciences, Prague, Czech Republic. [14] Molecular Endocrinology Laboratory (KMEB), Department of Endocrinology, Odense University Hospital, Odense C DK-5000, Denmark. ✉email: michaela.tencerova@fgu.cas.cz

Obesity is a worldwide health problem associated with metabolic complications affecting whole body metabolism[1]. The higher accumulation of fat in the bones is one of the metabolic complications leading to high fracture risk[2]. Obesity is mainly caused by excess caloric intake or an unbalanced diet, usually consisting of saturated and *trans*-unsaturated fatty acids (FAs) linked with higher body mass index (BMI) and complications affecting bone and fat metabolism[3]. Previous studies, including our findings, documented[4–7] that obesity negatively affects bone quality and phenotype of bone marrow stromal cells (BMSCs) due to increased adipogenesis and senescent microenvironment, which leads to higher bone fragility. Treatment of obesity, type 2 diabetes, and bone fracture risk is challenging for healthcare systems. It usually includes lifestyle changes such as dietary intervention or increased physical activity (reviewed in[8,9]). Omega-3 polyunsaturated fatty acids (omega-3 PUFAs), namely docosahexaenoic acid (DHA; 22:6n-3) and eicosapentaenoic acid (EPA; 20:5n-3), are fatty acids (FAs) that cannot be synthesized in sufficient amounts in the body. Therefore, they represent essential diet components with multiple beneficial effects on health (reviewed in[10]). EPA and DHA play a role as natural hypolipidemics, which reduce the accumulation of hepatic fat[11], ameliorate low-grade inflammation caused by obesity[12], increase adiponectin plasma levels[13] and enhance intestinal FA oxidation[14,15]. Moreover, several animal[16,17] and human[18] studies have also shown a positive effect of omega-3 PUFAs on bone health in different pathological conditions, including osteoporosis, obesity, or aging. However, the effect of omega-3 PUFAs on BMSC metabolism and bone marrow adiposity (BMA) under obesogenic conditions has not yet been thoroughly investigated.

Recent animal studies using high fat diet (HFD) supplemented with omega-3 PUFAs in aging osteoporotic models showed a diminished negative impact on bone loss, osteoclastogenesis and BMA[19,20]. Further, Cao et al.[21] reported positive effects of omega-3 PUFAs supplementation on bone parameters in HFD-fed C57BL/6 male mice over 6 months of dietary intervention. However, these studies did not focus their investigation on the early timepoint of dietary intervention with omega-3 PUFAs in younger animals, along with obesogenic conditions corresponding to the early impact of metabolic disturbances on bone homeostasis and stem cell properties in the bone microenvironment.

Previous in vitro studies documented that omega-3 PUFAs induce changes in BMSC plasma membrane composition, which leads to higher osteoblastic differentiation of these cells[22]. In addition, a study by Cugno et al.[23] showed that omega-3 PUFA supplementation of C3H10T1/2 cells could restore its osteoblastic differentiation capacity and inhibit the up-regulation of osteoclast (OC) differentiation of RAW264.7 cells. However, the exact molecular mechanism behind the omega-3 PUFA effect on bone cells and BMA in vivo is unknown.

Thus, the present study aimed to investigate whether omega-3 PUFA supplementation in HFD diet may prevent a detrimental effect on bone microstructure, bone marrow adipose tissue (BMAT), and molecular characteristics of BMSCs and OCs in an animal model of diet-induced obesity using young C57BL/6N male mice.

## Results

**Omega-3 PUFAs improve bone parameters in obese mice.** To investigate the effect of omega-3 PUFAs on metabolic and bone parameters in a HFD-induced animal model of obesity, 12-week-old C57BL/6N male mice were randomly assigned to 8-week-feeding with HFD or HFD supplemented with omega-3 PUFAs (HFD + F) and chow diet (ND) as a control. As a result,

metabolic parameters, including body weight gain, glucose tolerance, and fasting insulinemia were worsened by HFD as shown in our previous studies[6,24], which were improved in HFD + F compared to the HFD group without any effect on food intake (Supplementary Fig. 1a–e).

As an extension of the interventional studies mentioned above[6,24], the impact of omega-3 PUFA supplementation in HFD on bone physiology was investigated. µCT analyses of the proximal tibia and L5 vertebrae were performed in treated mice. HFD did not affect bone microstructure in the proximal tibia compared to the ND group (Fig. 1a, b; Supplementary Fig. 2a–d), while omega-3 PUFA supplementation (i.e., HFD + F diet) increased trabecular bone volume (Tb.BV/TV) and trabecular number (Tb.N) in proximal tibia compared to the HFD group (Fig. 1a, b) with no change in other cortical and trabecular parameters measured in the tibia (Supplementary Fig. 2a–d). Further, in L5 vertebrae, trabecular number (Tb.N), trabecular thickness (Tb.Th.) and trabecular separation (Tb.Sp) were not changed in treated mice (Supplementary Fig. 2e–g). However, HFD increased L5 cortical porosity (Ct.Po.) and decreased cortical thickness (Ct.Th.), when compared to the ND group (Fig. 1c, d), while omega-3 PUFA-enriched diet improved L5 Ct.Po (HFD vs. HFD + F, $p = 0.0101$), Ct.Th. (HFD vs. HFD + F, $p = 0.0384$) and cortical bone volume (Ct.BV/TV) (HFD vs. HFD + F, $p = 0.0465$) and cortical area fraction (B.Ar/T.Ar) (HFD vs. HFD + F, $p = 0.0465$) (Fig. 1c–f) compared to the HFD group, reaching more improved results than ND group (Fig. 1c, d). These changes are depicted in the 3D representative pictures of cortical bone with highlighted pores in the cortex (Fig. 1g). Further, measurement of mechanical properties using a three-point bending test showed stronger femora in HFD + F mice compared to HFD and ND group (Fig. 1h). Moreover, assessment of the ratio of circulating levels of bone formation marker procollagen type 1 N-terminal propeptide (P1NP) and bone resorption marker tartrate-resistant acid phosphatase (TRAP) (P1NP/TRAP) revealed a increased bone formation rate in the HFD + F vs. ND group (ND vs. HFD + F, $p = 0.0038$) (Fig. 1i; Supplementary Fig. 2h, i).

Thus, these findings demonstrated that omega-3 PUFAs supplemented for 8 weeks diminished the negative impact of HFD feeding on bone microstructure and bone strength in obese mice.

**Omega-3 PUFAs decrease BMAT volume in obese mice.** To further examine the impact of omega-3 PUFA treatment on BMA, BMAT volume was analyzed using contrast-enhanced X-ray microfocus computed tomography (CECT). In order to visualize BMAT in mouse bones, we used a contrast-enhancing staining agent (CESA), Hexabrix, which has been previously shown to visualize the adipocytes in the bone marrow (BM) cavity[6] and allowed faster tissue staining as it is a smaller molecule compared to other used CESA 1:2 Hafnium-substituted Wells-Dawson polyoxometalate (POM)[25].

CECT analysis of Hexabrix-stained bone marrow adipocytes (BMAds) showed increased BMAd number in HFD compared to the ND group, as previously published using $OsO_4$[4,5]. This increase was prevented by omega-3 PUFAs in HFD + F mice (HFD vs. HFD + F, $p = 0.0005$) (Fig. 2a, b). Further image analysis revealed that the differences in BMAd number across the three groups correlated with both BMAT volume (Fig. 2c) and diameter of adipocytes (Fig. 2d). Interestingly, in response to omega-3 PUFAs, both parameters were normalized to the level of the ND group. These changes in BMAT volume measured by CECT were also confirmed by histomorphometric analysis of H&E-stained sections of the proximal tibia (Fig. 2e–g).

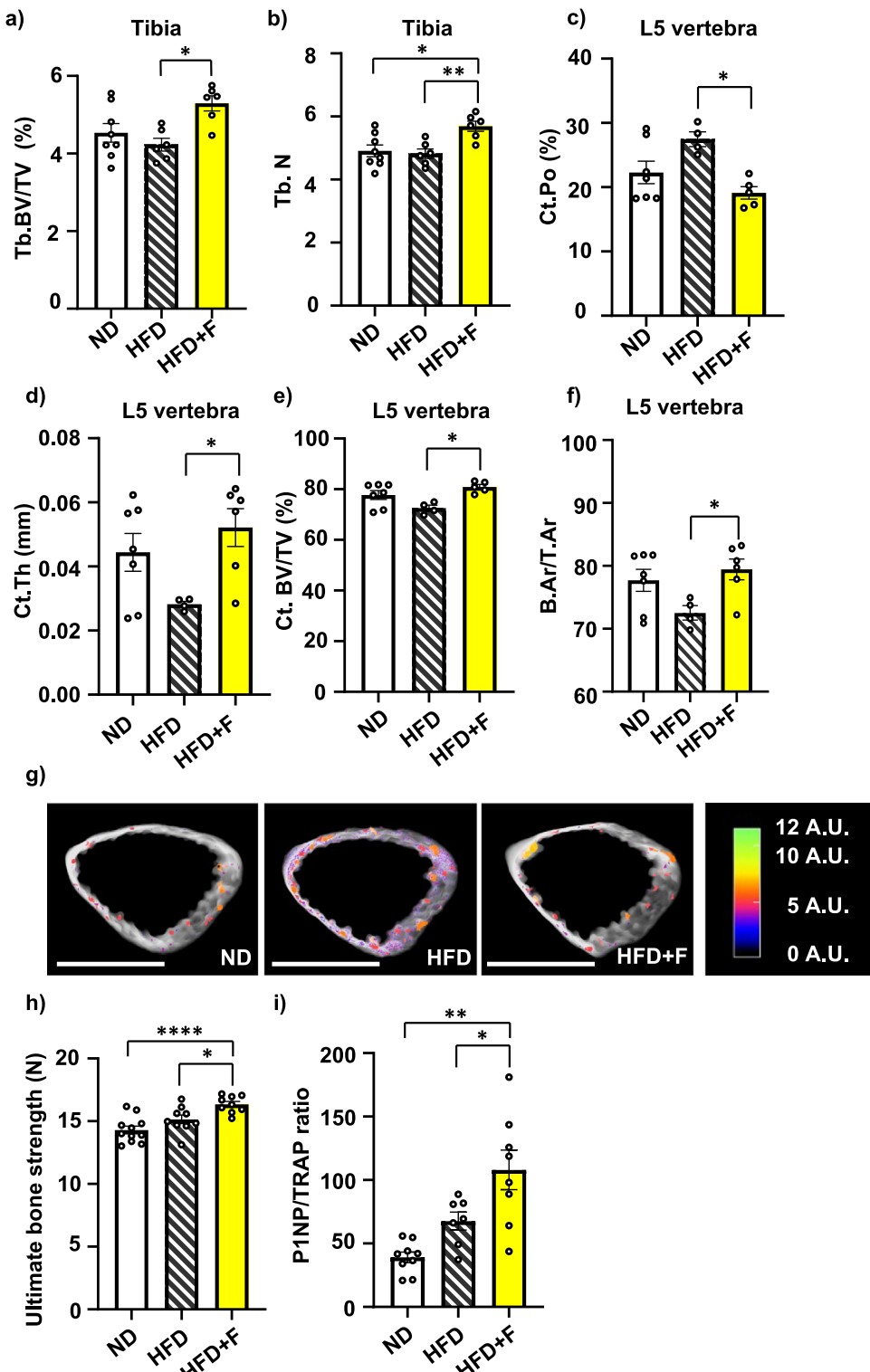

In summary, this data shows the impressive impact of omega-3 PUFAs in terms of BMAT volume reduction in the bones of obese mice, which reached the values of ND mice.

**Omega-3 PUFAs improve the lipidome and metabolome of circulating plasma and BM in obese mice.** To further characterize the impact of omega-3 PUFAs on plasma levels and bone microenvironment in obese mice, a global lipidomic and metabolomic analysis was conducted using liquid chromatography-mass spectrometry (LC-MS) on samples from plasma, BM, and bone powder (BP) obtained from treated mice. The analysis revealed a total of 905 metabolites, including polar metabolites and several simple and complex lipids across these three biological matrices with an overlap of 83%. Furthermore, the dataset was subjected to partial least-squares discriminant analysis (PLS-DA). The PLS-DA results demonstrated a distinct separation of HFD and HFD + F groups from ND in all matrices based on individual lipid species, while less separation using polar metabolites was observed (Supplementary Fig. 3a–c).

**Fig. 1 Omega-3 PUFAs protect bones from the detrimental effect of HFD in obese mice. a, b** µCT analysis of trabecular bone in proximal tibia in treated mice. Trabecular parameters were calculated as (**a**) trabecular bone volume per total volume of trabecular space (Tb.BV/TV) and (**b**) trabecular number (Tb.N). µCT analysis of cortical parameters of L5 vertebrae was calculated as (**c**) relative amount of pore space in the volume of cortical bone (cortical porosity (Ct. Po)), (**d**) mean cortical thickness (Ct. Th), and (**e**) cortical bone volume per total volume (Ct. BV/TV). **f** cortical area fraction (B.Ar/T.Ar) (n = 4-8 per group; one-way ANOVA, Tukey's multiple comparison test with *$p \leq 0.05$, **$p \leq 0.01$). **g** Representative pictures of 3D reconstructed µCT images from cortical bone analysis of L5 vertebrae (scale bar 1000 µm) with the colorimetric scale of pore size (scale bar 0–12 A.U.: shades of pseudocolors represent size of pores divided into categories according to their volume). Pictures were created using CTvox. **h** The ultimate bone strength of femurs was evaluated as the first point of the plateau of the load-displacement curve measured during a three-point bending test (n = 9–11 per group; one-way ANOVA, Tukey's multiple comparison test, with *$p \leq 0.05$, ****$p \leq 0.0001$). **i** Analysis of the ratio of circulating markers of bone resorption (P1NP) and bone formation (TRAP) in murine plasma samples after 8 weeks of dietary intervention (n = 8-10 per group; one-way ANOVA, Tukey's multiple comparison test with *$p \leq 0.05$, **$p \leq 0.01$). Data are presented as mean ± SEM (groups coding: white column-ND, black line shading-HFD, yellow column-HFD + F).

From a broader perspective, using the sum of the abundances of unique lipid species for each lipid class, 24 out of 32 were significantly decreased in plasma, especially triglycerides (TG), diacylglycerols (DG), cholesterol, ceramides (CER), and phosphatidylcholines (PC) in HFD + F compared to HFD group (Fig. 3a). On the other hand, only three lipid classes (lysophosphatidylglycerol (LPG), N-acyl ethanolamines (NAE), and ether-linked phosphatidylinositol (EtherPI)) were altered in BM (Fig. 3b), and four lipid classes (fatty acid ester of hydroxy fatty acid (FAHFA), EtherPI, ether-linked triacylglycerol (EtherTG), and cholesteryl ester (CE)) in BP (Fig. 3c). Further detailed analysis of the abundance of individual lipid species and polar metabolites revealed interesting changes in circulating metabolites in plasma compared to BM and BP (Supplementary Fig. 4, 5, 6). Specifically, the significant enrichment of DHA (22:6n-3) and EPA (20:5n-3) in both plasma and BM in the HFD + F group vs. HFD group (Supplementary Fig. 4, 5) suggests their contribution to the beneficial effect on improving metabolism and local BM microenvironment in obese mice. Furthermore, these findings showed that dietary supplementation of omega-3 PUFAs in HFD has a positive impact on decreasing circulating levels of lipids in HFD mice along with increased levels of DHA and EPA in BM, thus confirming data from µCT evaluation on different BMAT compositions in individual groups.

**Omega-3 PUFAs improve the differentiation capacity of BMSCs in obese mice.** To evaluate the impact of omega-3 PUFAs treatment on cellular characteristics of BMSCs, primary BMSCs were isolated from mice subjected to different dietary interventions to evaluate their molecular phenotype. While a short-term proliferation rate of primary cultures did not differ between the groups (Supplementary Fig. 7a), the colony-forming units-fibroblast (CFU-f), a marker of stem cell properties, showed an increased number in the HFD + F compared to the HFD group (Fig. 4a, b).

Further, HFD increased adipocyte (AD) differentiation of primary BMSCs after 8-week treatment in mice compared to cells isolated from ND mice, which was visualized by Oil Red O (ORO) staining (Fig. 4c). This impact of HFD on BMSCs was prevented by omega-3 PUFA supplementation as primary cultures of BMSCs from HFD + F mice showed less ORO-positive AD (Fig. 4c). These changes were also confirmed by gene expression profile of adipogenic markers (*Pparγ2, Cebpa, Adipoq, Cd36, Fsp27, Vcam*), which were decreased in BMSCs from the HFD + F compared to HFD or ND group (Fig. 4d). Moreover, expression of oxidative stress (*Hmox1*) and inflammatory markers (*TNFα*) was decreased in the HFD + F compared to the HFD group, and the expression of senescence marker (*p53*) was downregulated in HFD + F compared to ND (Fig. 4e).

On the other hand, the osteoblast (OB) differentiation potential of primary BMSCs was decreased by HFD measured by Alizarin staining (AZR) and alkaline phosphatase (ALP) activity in comparison to the ND group, which was improved by supplementation with omega-3 PUFAs in obese mice (Fig. 4f±h). Importantly, these changes were further confirmed by gene expression of osteoblastic genes (*Alpl, Oc, Col1a1, Bmp2, Ctnnb1*) (Fig. 4i), which is consistent with the results from µCT analysis of bone and BMAT parameters. Thus, these data demonstrate that omega-3 PUFA supplementation of HFD positively impacts the cellular characteristics of BMSCs and improves their differentiation capacity.

**Omega-3 PUFAs affect the differentiation capacity of OCs in obese mice.** Previous studies documented an inhibitory effect of omega-3 PUFAs on OC differentiation in vitro and in vivo using OVX or aging mouse model[23,26,27]. Thus, we evaluated OC differentiation capacity with omega-3 PUFA treatment in vitro and in primary BM cells isolated from treated mice after 8 weeks of diet.

In vitro treatment of BM derived cells with 100 µM EPA, DHA or mix EPA and DHA for 5 days induced inhibition of OC differentiation compared to non-treated OCs, which was also confirmed by gene expression of osteoclastic genes (*Trap* and *Ctsk*) (Supplementary Fig. 8a–c).

In primary BM cells isolated from treated mice, TRAP staining revealed increased number of total TRAP + OCs in HFD compared to ND group (ND vs. HFD, $p = 0.0258$), while omega-3 PUFA supplementation decreased OC formation (Fig. 5a, d). Further, we found increased formation of large OCs (cell diameter above 500 µm, more than 30 nuclei) in HFD compared to ND group (ND vs. HFD, $p = 0.038$), which was decreased in HFD + F group (HFD vs. HFD + F, $p = 0.048$) (Fig. 5b, d). However, TRAcP activity measured in primary differentiated OCs showed no differences among groups regardless of the diet (Fig. 5c), which was also confirmed by gene expression of osteoclastic genes (*Trap, Ctsk*) (Fig. 5e). However, there was a trend towards decreased gene expression of inflammatory gene *Il1b* (Fig. 5f) in HFD + F compared to HFD group suggesting less inflammatory profile of OCs with lower OC differentiation induced by omega-3 PUFA supplementation in diet.

**Omega-3 PUFAs improve insulin and inflammatory responsiveness in HSCs in obese mice.** Previous studies applying omega-3 PUFAs in obese mice showed their positive impact on inflammation and insulin signaling pathways in peripheral adipose tissue[24]. Thus, we evaluated AKT and NFκB signaling pathways in HSCs, progenitors of immune cells, derived from treated mice.

Insulin stimulation measured by AKT phosphorylation (pAKT S473/total AKT, and pAKT T308/total AKT) showed that insulin

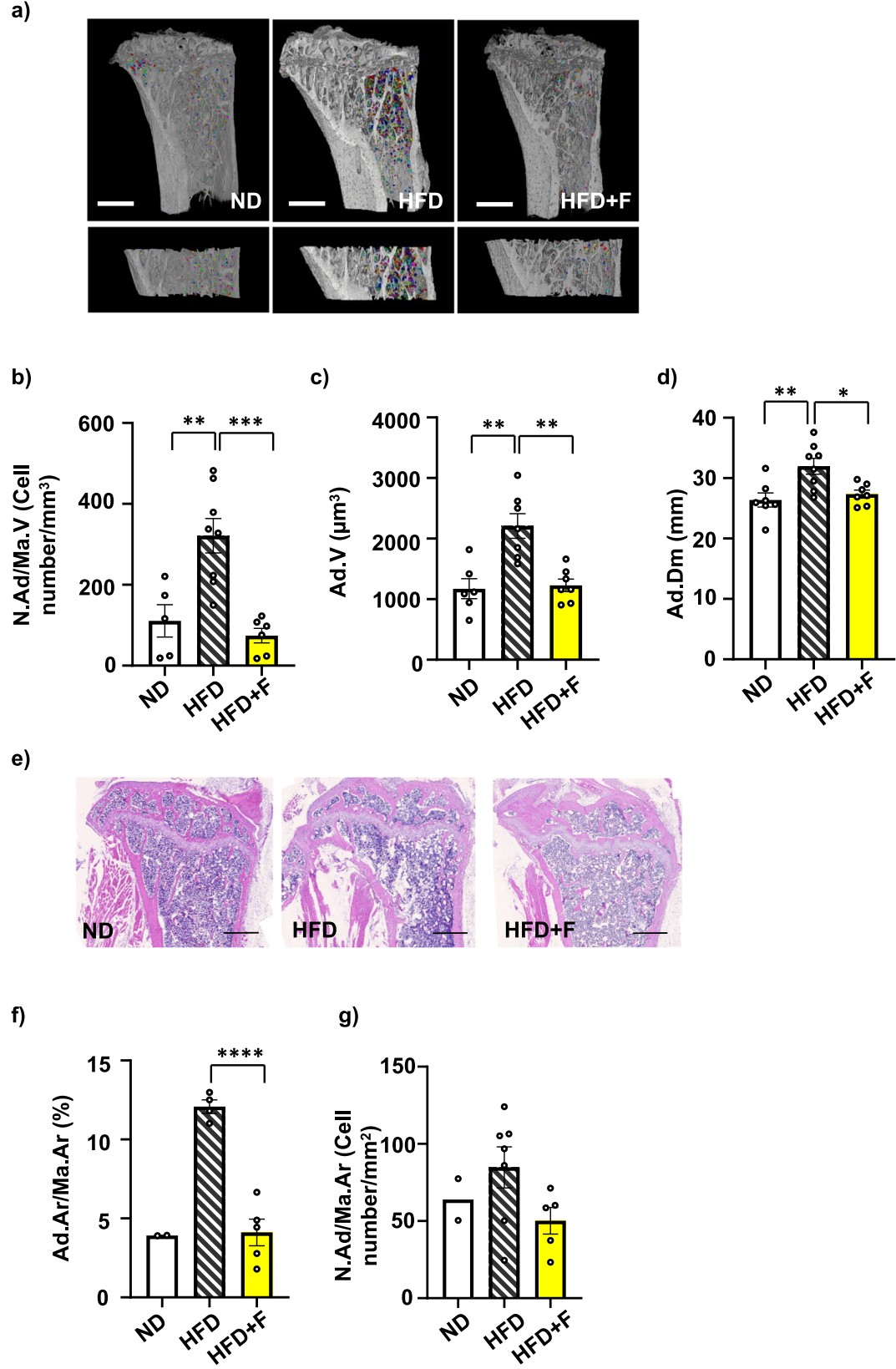

signaling in primary HSCs of HFD mice was not impaired when compared to ND cells (Fig. 6a–d; Supplementary Fig. 9). Interestingly, HFD + F HSCs showed decreased insulin responsiveness compared to HFD HSCs, especially for pAKT T308/total AKT (HFD vs HFD + F, $p = 0.0341$) (Fig. 6c, d; Supplementary Fig. 9).

Further, inflammatory stimulation of primary HSCs by LPS induced activation of NFκB signaling measured by p65 phosphorylation (p-p65/totalp65) in the HFD group, which was decreased in the HFD + F group (HFD vs. HFD + F, $p = 0.0125$) (Fig. 6e, f; Supplementary Fig. 10). Moreover, there was a trend towards decreased expression of bone resorption markers

**Fig. 2 Omega-3 PUFAs decrease bone marrow adiposity in obese mice. a** Representative pictures of bone marrow adipocytes (BMAds) stained with contrast agent Hexabrix® in the whole proximal tibia and zoomed pictures of BMAds in the selected region of interest in proximal tibia (defined in Material and Methods) of contrast enhanced samples scanned by μCT. Pictures were created using Avizo Software (version 2021.1, ThermoFisher, scale bar 1000 μm). **b** Quantification of BMAd density (N.Ad/Ma.VCell number/mm³)). **c** Evaluation of BMAT volume (Ad.V (μm³)) in the selected region of interest in the proximal tibia. (**d**) Quantitative evaluation of the diameter of Hexabrix-stained BMAds in the tibia (Ad.Dm (mm)). (n = 6-8 per group; one-way ANOVA, Tukey's multiple comparison test with $*p \leq 0.05$, $**p \leq 0.01$, $***p \leq 0.001$). Histomorphometric evaluation of the BMAds showing (**e**) representative pictures from H&E staining of adipocytes from a histological section of the proximal tibia from mice fed with HFD supplemented with omega-3 PUFAs (scale bar 500 μm), (**f**) total adipocyte area (Ad.Ar/Ma.Ar (%)), and (**g**) BMAd density affected by omega-3 supplementation in obese mice (N.Ad/Ma.Ar (Cell number/mm²)) (n = 2-7 per group; one-way ANOVA, Tukey's multiple comparison test with $***p \leq 0.001$, $****p \leq 0.0001$. Data are presented as mean ± SEM (groups coding: white column-ND, black line shading-HFD, yellow column-HFD + F).

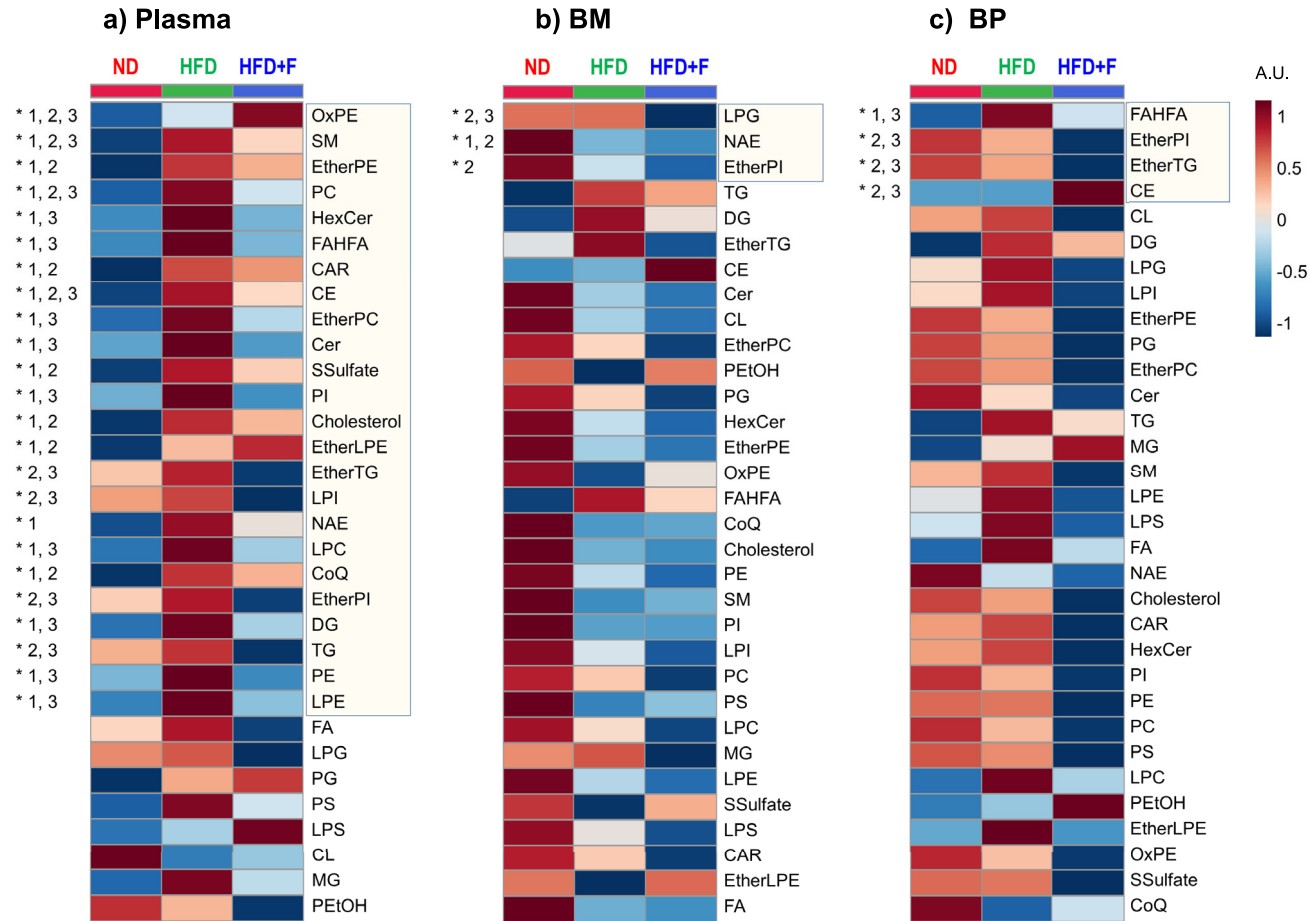

**Fig. 3 Lipidomic analysis revealed the effect of omega-3 PUFAs comparing circulating plasma, bone marrow, and bone powder in obese mice.** Global lipidomic profiling of plasma (PL), bone marrow (BM), and bone powder (BP) samples obtained from investigated mice using LC-MS. Heatmap of the sum of the abundances of unique lipid species for each lipid class for (**a**) plasma, (**b**) bone marrow, and (**c**) bone powder with group averages (n = 5-10 per group). Lipid classes statistically altered are marked by an asterisk (*) based on ANOVA with p(FDR) < 0.05. Differences between groups are marked as 1 (ND vs. HFD; $p < 0.05$), 2 (ND vs. HFD + F; $p < 0.05$), and 3 (HFD vs. HFD + F; $p < 0.05$) (gradient color keys used in normalized intensity, A.U.). (Lipid class annotation: CAR acylcarnitine, CE cholesteryl ester, CL cardiolipin, Ce, ceramide, CoQ coenzyme Q, DG diacylglycerol, EtherPE ether-linked phosphatidylethanolamine, EtherLPE ether-linked lysophosphatidylethanolamine, EtherPI ether-linked phosphatidylinositol, EtherPC ether-linked phosphatidylcholine, EtherTG ether-linked triacylglycerol, FA free fatty acid, FAHFA fatty acid ester of hydroxy fatty acid, HexCer hexosylceramide, LPC lysophophatidylcholine, LPE lysophosphatidylethanolamine, LPG lysophosphatidylglycero, LPI lysophosphatidylinositol, LPS lysophosphatidylserine, MG monoacylglycerol, NAE N-acyl ethanolamines, OxPE oxidized phosphatidylethanolamine, PC phosphatidylcholine, PE phosphatidylethanolamine, PEtOH phosphatidylethanol, PG phosphatidylglycerol, PI phosphatidylinositol, PS phosphatidylserine, SM sphingomyelin, SSulfate sterol sulfate, TG triacylglycerol).

(*Rankl, Opg*) in HSCs of the HFD + F vs. HFD group (Fig. 6g). These findings suggest that omega-3 PUFAs decreased insulin and inflammatory signaling in progenitors of immune cells under obesogenic conditions.

**Omega-3 PUFAs slow down metabolic activity along with decreased senescence in BMSCs in obese mice.** To further evaluate the effect of omega-3 PUFAs on cellular metabolism, we measured the bioenergetic profile in primary BMSCs isolated

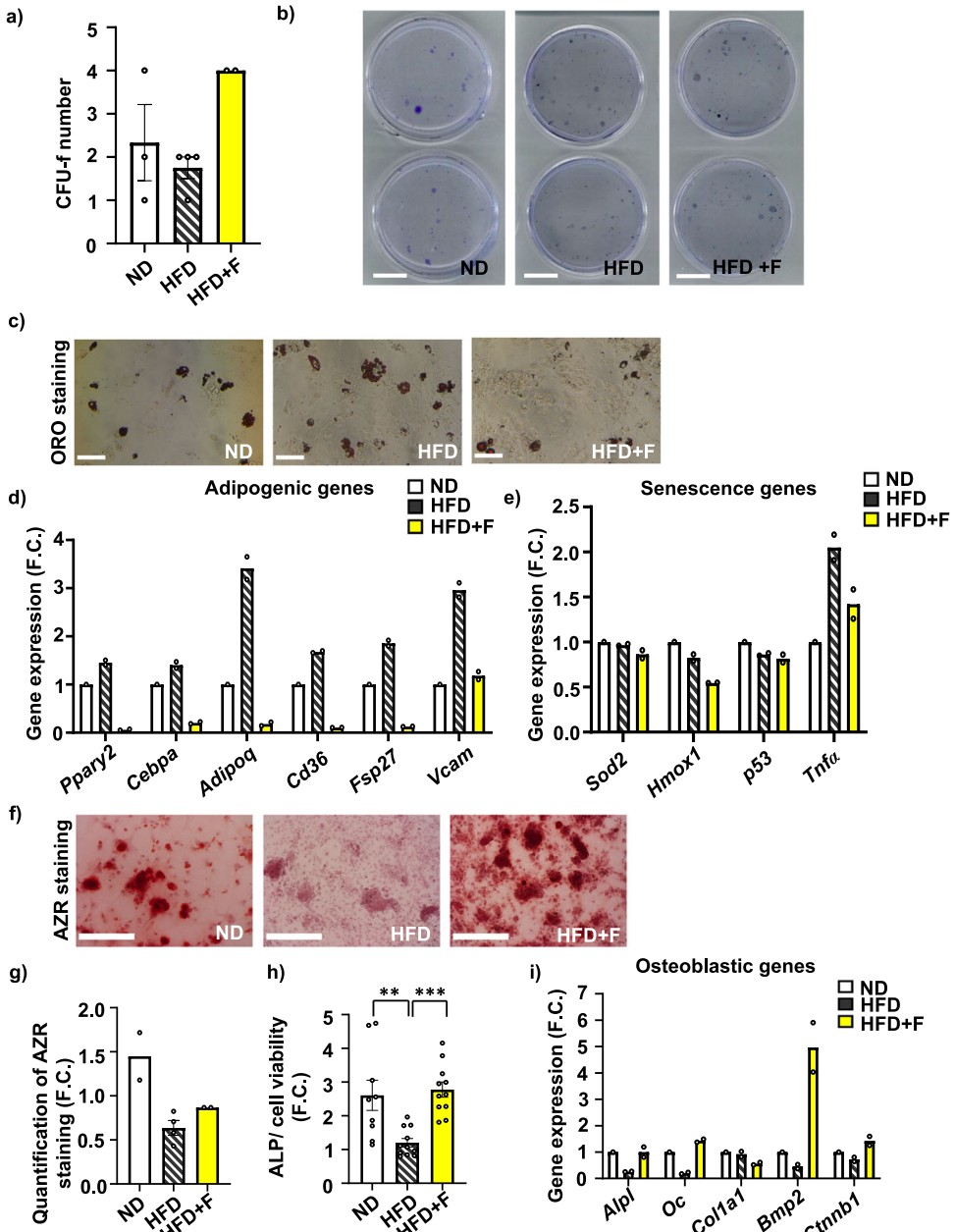

**Fig. 4 Effect of omega-3 PUFAs on stem cell properties of primary BMSCs.** Analysis of stem cell phenotype by (**a**) colony-forming units-fibroblast (CFU-f) analysis of freshly isolated BMSCs with (**b**) representative pictures of CFU-f analysis (scale bar 20 mm). (n = 2–4 from pooled samples; t test *p ≤ 0.05: HFD vs HFD + F) Data are presented as mean values ± SEM. **c** Representative pictures of Oil red O (ORO) stained lipid droplets in AD differentiated BMSCs (scale bar 500 μm; 10x magnification). **d**, **e** Gene expression profile of BMSCs differentiated towards adipocytes in D10. **d** Gene expression of adipogenic genes (*Pparγ2, Cebpa, Adipoq, Cd36, Fsp27, Vcam*) and (**e**) oxidative stress, senescence and inflammatory genes (*Sod2, Hmox1, p53, Tnfα*) (n = 2 from pooled samples). Data are presented as mean fold change (F.C.) of gene expression normalized to BMSCs from ND group ± SEM. **f** Representative pictures of Alizarin Red (AZR) staining for calcified matrix mineralization of OB differentiated BMSCs (scale bar 500 μm; 10x magnification) and (**g**) quantification of eluted AZR staining of mineralized matrix in OB differentiated BMSCs in D10. (n = 2–4; one-way ANOVA, Tukey's multiple comparison test with *p ≤ 0.05. Data are presented as mean fold change (F.C.) of A$_{500}$ in OB differentiated cells (OB) normalized to BMSCs from ND group ± SEM. **h** Measurement of alkaline phosphatase (ALP) activity normalized to cell viability in OB differentiated BMSCs in D7. (n = 9–11; one-way ANOVA, Tukey's multiple comparison test with **p ≤ 0.01, ***p ≤ 0.001). Data are presented as mean fold change of ALP activity in OB differentiated cells (OB) from each experimental group ± SEM. **i** Gene expression profile of osteoblastic markers (*Alpl, Oc, Col1a1, Bmp2, Ctnnb1*) in OB differentiated BMSCs in D10 (n = 2 from pooled samples). Data are presented as mean fold change (F.C.) of gene expression normalized to BMSCs from ND group ± SEM (groups coding: white column-ND, black line shading-HFD, yellow column-HFD + F).

from mice at the end of dietary interventions in their undifferentiated state. Using Seahorse technology, measurement of cellular respiration in primary BMSCs showed stimulation of basal respiration and higher maximal respiration rate induced by carbonyl cyanide-4-(trifluoromethoxy) phenylhydrazone (FCCP) in

HFD BMSCs compared to ND BMSCs (Fig. 7a). Interestingly, this HFD-induced increased OCR activity was diminished in BMSCs from HFD + F mice. Similarly, glycolysis measurement revealed that HFD + F treatment decreased the glycolytic capacity of the corresponding BMSCs compared to cells from ND or

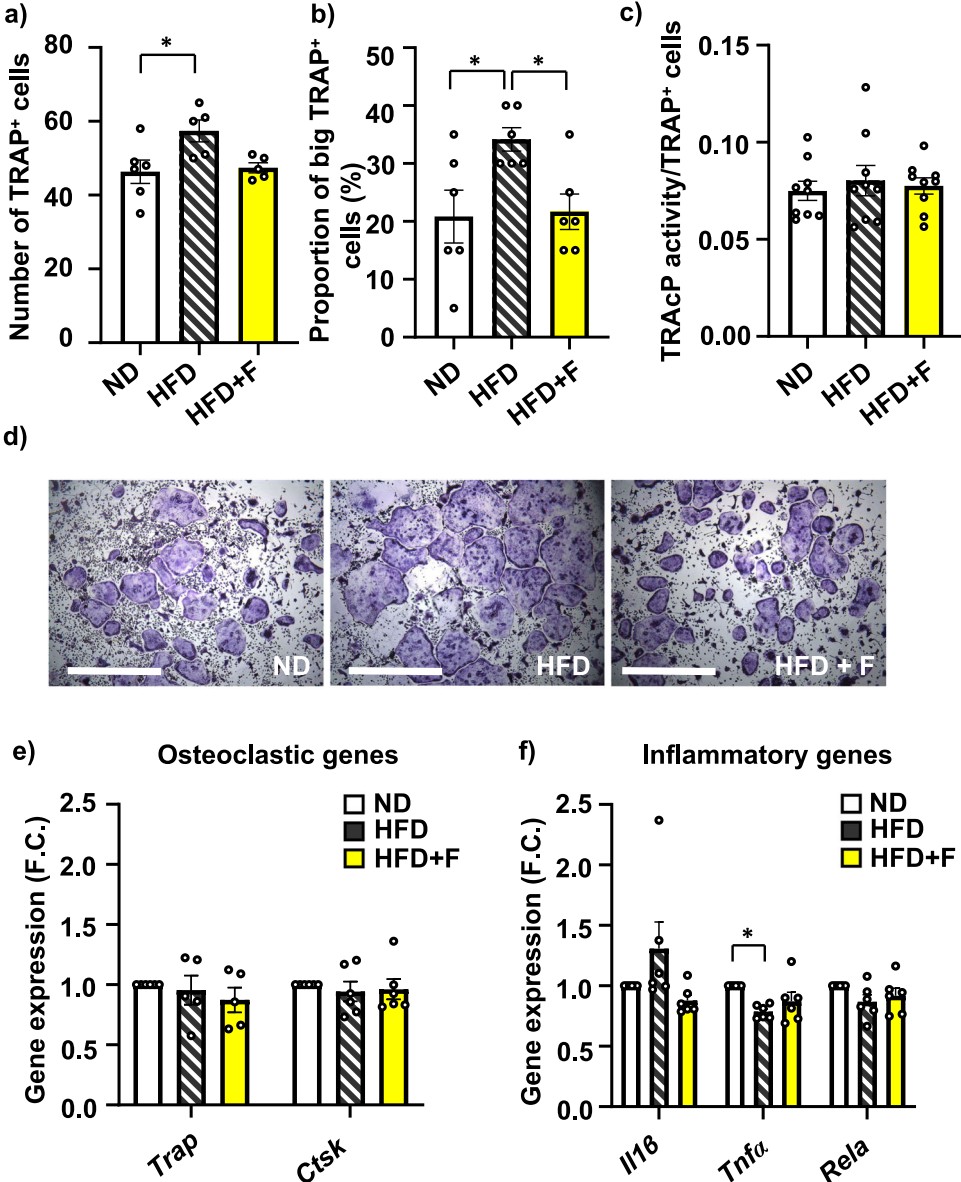

**Fig. 5 Omega-3 PUFAs affect the differentiation capacity of OCs in obese mice. a–c** Analysis of osteoclast differentiation from BM cells isolated from treated mice. **a** Number of TRAP⁺ cells, (**b**) Proportion of large TRAP⁺ OCs from total OCs (cell diameter above 500 µm) and (**c**) measurement of TRAcP activity normalized to number of TRAP⁺ cells (n = 5–6 per group; one-way ANOVA, Tukey's multiple comparison test with *$p \leq 0.05$) Data are presented as mean ± SEM. **d** Representative pictures of TRAP⁺ cells after 5 days of differentiation. Gene expression analysis of (**e**) oscteoclastic genes (*Trap, Ctsk*) and (**f**) inflammatory genes (*Il1β, Tnfα, Rela*) measured in OC differentiated cells after 5 days of differentiation. (n = 5–6 mice per group; one-way ANOVA, Tukey's multiple comparison test with * $p \leq 0.05$). Data are presented as mean fold change (F.C.) of gene expression normalized to ND group ± SEM (groups coding: white column-ND, black line shading-HFD, yellow column-HFD + F).

HFD mice (Fig. 7b). This is well documented in the energy phenotype profile under basal conditions of BMSCs from HFD + F group showing clear shift towards a quiescent metabolic state, contrasting with rather energetic profile of BMSCs from the HFD and ND groups (Fig. 7c). Further, measurement of β-gal activity and reactive oxygen species (ROS) production in primary BMSCs showed decreased senescent phenotype in the HFD + F compared to the HFD group (HFD vs. HFD + F, $p = 0.0185$; $p = 0.0223$, respectively) (Fig. 7d, e) accompanied by decreased gene expression of senescence-associated secretory phenotype (SASP) markers (e.g., *Fas, Fasgl, Il1β, Vegfa, Vcam, Il10, Il1rn*) in primary BMSCs (Fig. 7f) and HSCs from the HFD + F compared to HFD group (Supplementary Fig. 7b).

These findings demonstrate that omega-3 PUFA treatment in mice slows down the negative effect of HFD on cellular metabolism and senescence in BMSCs and HSCs, thus highlighting their beneficial impact on the BM microenvironment.

## Discussion

Non-pharmacological treatments, including dietary restriction and physical activity have been shown to improve metabolic and bone parameters in metabolic diseases such as obesity and osteoporosis. Animal and clinical studies demonstrated that consumption of omega-3 PUFAs can impact bone health by modulating calcium metabolism, prostaglandin synthesis, lipid

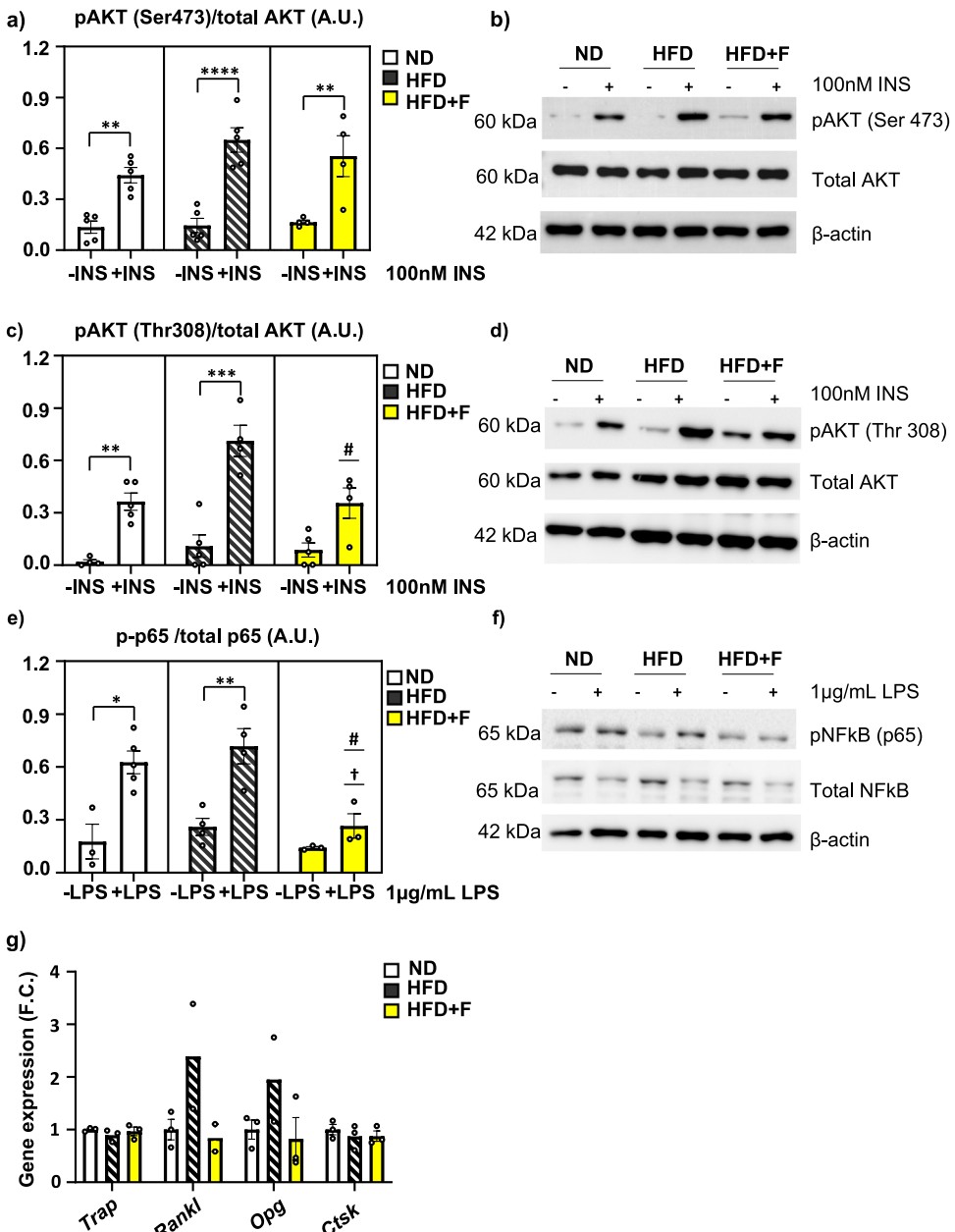

**Fig. 6 Omega-3 PUFAs affect insulin and inflammatory signaling and senescence in HSCs of obese mice. a** Densitometry evaluation of western blot images representing the results of insulin stimulation (100 nM, 15 min) of p-S473-AKT/total AKT in primary HSCs from the ND, HFD and HFD + F groups (n = 4–5 per group) and (**b**) representative western blot images; (**c**) Densitometry evaluation of western blot images representing the results of insulin stimulation (100 nM, 15 min) of p-T308-AKT/total AKT in primary HSCs from the ND, HFD and HFD + F groups (n = 4–5 per group) and (**d**) representative western blot images. **e** Densitometry evaluation of western blot images representing the results of lipopolysaccharide (LPS) stimulation (1 μg/ml, 15 min) of p-NFκB/totalNFκB in primary HSCs from the ND, HFD and HFD + F groups (n = 3–5 per group) and (**f**) representative western blot images. Data are presented as mean densitometry ± SEM, t test, *$p ≤ 0.05$, **$p ≤ 0.01$, ***$p ≤ 0.001$, ****$p ≤ 0.0001$ significant difference between –INS vs. +INS, one-way ANOVA, Tukey's multiple comparison test, †: ND vs. other +INS groups with $p ≤ 0.05$, #: HFD vs. other +INS groups with $p ≤ 0.05$. **g** Gene expression of bone resorption genes (*Trap, Rankl, Opg, Ctsk*) in primary HSCs (n = 3 per group from pooled samples, one-way ANOVA, Tukey's multiple comparison test. Data are presented as mean fold change (F.C.) of gene expression normalized to HSCs from ND ± SEM (groups coding: white column-ND, black line shading-HFD, yellow column-HFD + F).

oxidation, osteoblast formation, and osteoclastogenesis[28]. However, the underlying mechanisms of how omega-3 PUFAs modulate bone homeostasis in obesity-induced impairment of bone formation are not well documented. In the present study, we report that supplementation with omega-3 PUFAs improved metabolic and bone parameters combined with decreased BMA in obese mice, which were accompanied by increased OB differentiation potential, lower OC differentiation and lowered

senescent markers in primary BMSCs. These data support the beneficial effect of omega-3 PUFAs on bone and cellular metabolism of BMSCs and their potential use in the treatment of metabolic bone diseases.

In our study, 8-week-dietary intervention was used to characterize the effect of HFD supplementation with omega-3 PUFAs (enriched with DHA and EPA) on the adverse effects of obesity on bone and fat metabolism in 12-week-old C57BL/6 N male mice.

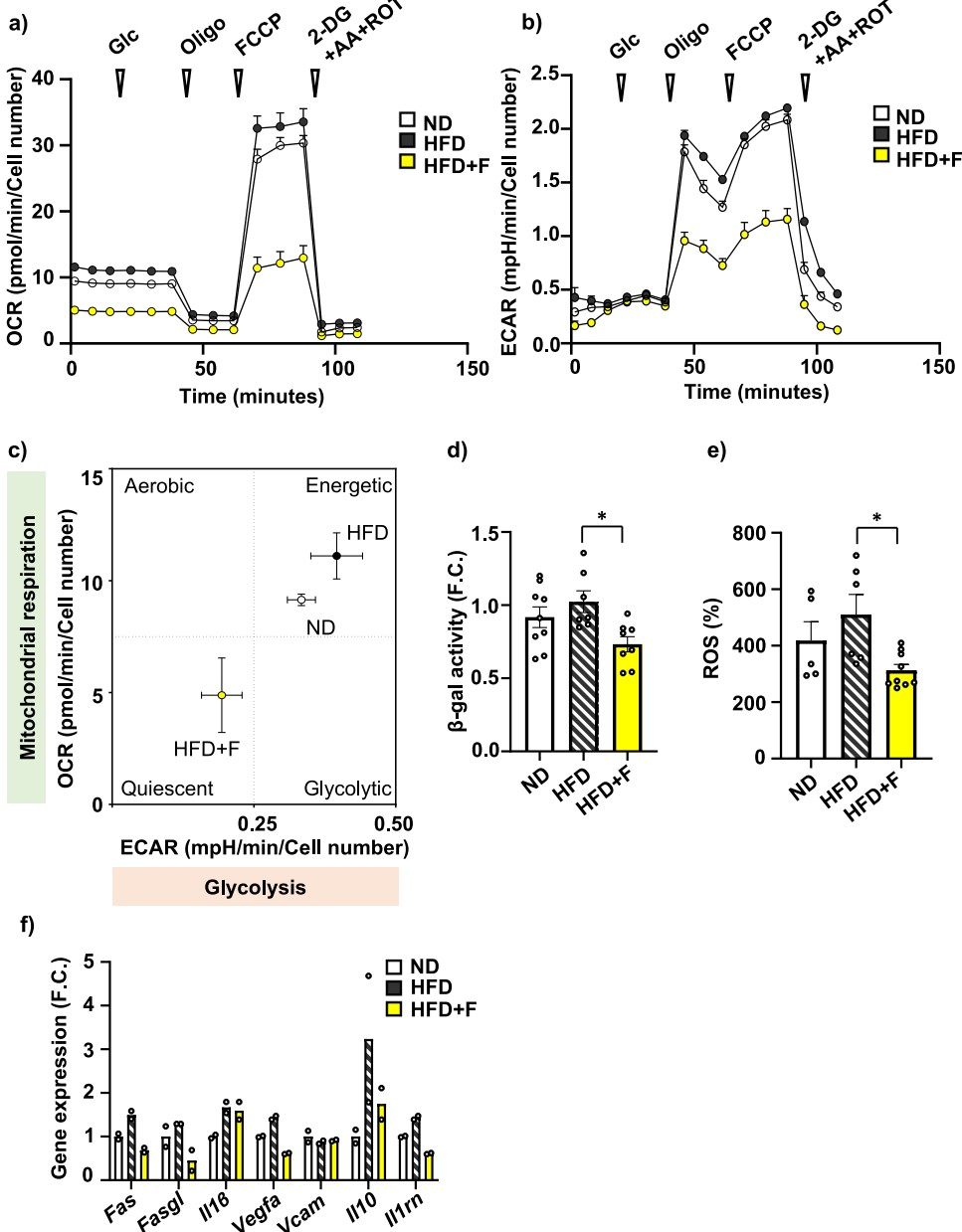

**Fig. 7 Omega-3 PUFAs decrease cellular metabolism in BMSCs of obese mice. a** Measurement of oxygen consumption rate (OCR) of primary BMSCs isolated from treated mice (n = 2 independent experiments with five replicates per group). **b** Measurement of extracellular acidification rate (ECAR) of primary BMSCs (n = 7–11 per group). Data are presented as mean ± SEM (groups coding: white circle-ND, black circle-HFD, yellow circle-HFD + F), (**c**) XF Energy map visualizing metabolic phenotype profile of cultivated BMSCs. The Energy map was calculated from basal OCR and ECAR mean values (n = 7–11 per group). **d** Analysis of senescence of cultivated mouse BMSCs using β-galactosidase activity measurement. **e** Measurement of ROS production in cultivated mouse BMSCs (n = 7–9 per group, one-way ANOVA, Tukey's multiple comparison test with * $p ≤ 0.05$). Data are presented as mean ± SEM. **f** Gene expression analysis of SASP markers in primary BMSCs obtained from treated mice exposed to 50 μM $H_2O_2$ for 6 h (*Fas, Fasgl, Il1β, Vegfa, Il10, Il1rn*) (n = 2 per group from pooled samples). Data are presented as mean fold change (F.C.) of gene expression normalized to ND group ± SEM (groups coding: white column-ND, black line shading-HFD, yellow column-HFD + F).

Previous animal studies investigating treatment with omega-3 PUFAs documented different results on bone phenotype, depending on the investigated conditions (growth, osteoporosis, aging, obesity, diabetes), sex (males/females), length of the treatment, and composition of the diet. Farahnak et al.[29] using younger growing animals reported that a 1% DHA-enriched diet increased lumbar bone density and cortical bone volume in 6-week-old female Sprague-Dawley rats after 10 weeks of treatment, which aligns with our study showing improved trabecular and cortical parameters in the proximal tibia and L5 vertebrae measured by

μCT, and enhanced bone strength, after 12 weeks of omega-3 PUFA supplementation in male mice. However, similar treatment in 3-week-old C57BL/6 J male and female mice[28] or 6-week-old male diabetic Zucker rats[30] did not induce changes in bone phenotype measured by dual-energy X-ray absorptiometry (DXA), after 9 weeks of treatment, which could be explained by a less sensitive method for evaluation of bone phenotype and different animal model of metabolic diseases.

Other animal studies using longer treatment with omega-3 PUFAs (from twenty to twenty-four weeks of diet) showed a

consistent positive effect of omega-3 supplementation on bone phenotype, emphasizing its importance in diet[19,21,23]. However, in any of these animal studies, the authors did not investigate the effect of omega-3 PUFAs on BMA and primary cells isolated from treated animals.

Recently, Hassan et al., using a murine model of senile osteoporosis (SAMP8 knockout mice), reported that an omega-3 PUFA-enriched diet administered for 10 months prevented age-related bone loss by reducing BMAT expansion (as measured by µCT), thus highlighting the impact of BMAT expansion in the development of metabolic bone diseases[20]. These findings nicely correlate with our observation on reduced BMA induced by an omega-3 PUFA-enriched diet using a shorter intervention period of 2 months. In contrast to the study of Hassan et al., our study used CECT to visualize the BMAds, which allowed a more precise analysis of their structural parameters and distribution in the BM cavity using Hexabrix CESA[31]. Hexabrix is a smaller molecule compared to previously used POM CESA and does not interact with the vasculature inside the BM, allowing faster tissue sample staining[25]. In addition, it is an unharmful compound and is consequently preferred over osmium tetroxide ($OsO_4$), which is highly toxic[32]. Thus, this contrast agent staining of BMAds is better for detecting the beneficial omega-3 PUFA-induced effects on the size and the number of BMAds. Furthermore, applying CECT allows to choose non-subjective gray value thresholds for segmentation of BMAds.

We also evaluated the molecular phenotype of primary BMSCs isolated from treated mice, which has not been investigated in previous animal studies. We found that the administration of omega-3 PUFA-enriched HFD improved OB differentiation of BMSCs and decreased AD differentiation and senescent phenotype compared to HFD BMSCs. These changes were accompanied by decreased NFκB and insulin signaling in HSCs after omega-3 PUFA treatment compared to the HFD group, suggesting lower inflammatory response and metabolic activity associated with a decreased aging process in immune cells[33]. These findings from primary BM cells support results obtained in vitro using omega-3 PUFAs treatment in C3H10T1/2 and RAW264.7 cells, which showed the inhibition of RANKL and PPARγ signaling pathway[23,34–36]. Furthermore, Levental and colleagues reported that omega-3 PUFAs induced changes in the plasma membrane of BMSCs, which led to higher OB differentiation[22].

In terms of OC differentiation, we observed increased total OC number without any effect on TRAcP activity in HFD compared to ND confirming data from previous animal study using 3-month HFD treatment[26]. This effect was decreased by omega-3 PUFA supplementation in diet. Interestingly, HFD increased formation of bigger OCs compared to ND, which was diminished by omega-3 PUFAs supplementation supporting data from previous in vitro and in vivo studies on inhibitory effect of omega-3 PUFAs on osteoclastogenesis and thus a positive effect of omega-3 PUFAs on bone formation[23,26,27]. However, further studies using single cell RNA seq analysis of OCs might reveal more information about the differences in resorption activity of OCs depends on the number of nuclei or their size. Recent study by McDonald et al.[37] reported interesting phenomenon on OC recycling process via osteomorphs, which affects bone remodeling process. However, it is not known how obesity or diet intervention may affect this process. Future studies are needed to answer these questions.

In our study, we were also interested in the effect of omega-3 PUFAs on cellular metabolism in primary BMSCs, which might explain the local changes in the BM microenvironment. Interestingly, bioenergetic profiling revealed quiescent phenotype (low glycolytic and respiratory activity) in primary BMSCs obtained from HFD + F mice compared to HFD BMSCs with higher energetic profile, which is opposite to response in peripheral adipocytes with increased mitochondrial biogenesis after omega-3 PUFA treatment[38]. The metabolic changes in BMSCs were accompanied by decreased ROS production and senescent activity, suggesting that omega-3 PUFAs protect the BM microenvironment from the harmful effect of obesogenic conditions by reducing the metabolic activity of primary BMSCs, which might support maintenance of the stemness and less epigenetic changes in BMSCs[39].

These beneficial effects on bone metabolism were further confirmed by metabolomic analysis showing a higher concentration of DHA and EPA not only in plasma but also in BM and bone matrix of HFD + F compared to HFD mice, which was associated with a positive impact on bone quality and lower BMA and improved bone microstructural changes. Besides omega-3 PUFAs, omega-6 and omega-9 PUFAs in the diet may benefit bone parameters[20,40]. Thus, it is important to pay more attention to the composition of different PUFAs in experimental diets and their beneficial effects on bone. Further, lipidomic analysis showed decreased levels of prostaglandin in plasma and BM, which correlates with better bone formation also documented in previous studies[30,41,42]. Interestingly, the metabolomic data revealed the different responses to diet in plasma and BM microenvironments, with a strong effect of diet on plasmatic levels of metabolites in comparison to a milder effect in BM or BP metabolite profile under dietary intervention. The metabolomics and cellular data highlight the difference in BM metabolism compared to the periphery and its relative resistance to diet.

The present study is based on the use of several innovative approaches. Firstly, unlike previous studies investigating the effect of omega-3 PUFA treatment on bone, we conducted a comprehensive analysis encompassing both bone and BMAd parameters. Additionally, we thoroughly analyzed the cellular and molecular properties of primary BMSCs and OCs derived from the treated animals. Secondly, we used an interventional study with a preventive design of the treatment to investigate how omega-3 PUFAs impact bone health in obesity. Thirdly, we employed state-of-the-art methods to enhance our understanding of cellular metabolism in BMSCs obtained from the treated mice.

On the other hand, our study has some limitations. We used only 2 months of dietary treatment in C56BL/6 N male mice that did not manifest with a lot of changes in bone and senescent phenotype in BMSCs in HFD condition compared to ND. However, the beneficial effect of HFD + F was already pronounced in comparison to HFD condition. Further, we did not investigate the effects of omega-3 PUFAs in female mice, which is important for future lifestyle recommendations for women with osteoporosis-related metabolic complications.

Taken together, our interventional study using HFD-induced obesity in mice showed that omega-3 PUFA supplementation improved metabolic and bone parameters in obese mice, which was accompanied by reduced BMA and better differentiation potential, and lower senescence in BMSCs. Thus, these data highlight the beneficial effects of omega-3 PUFAs on bone and cellular metabolism and their potential use in the treatment of metabolic bone diseases.

## Methods

**Animals and dietary interventions**. We utilized 10-week-old male C57BL/6 N mice obtained from Charles River Laboratories, Sulzfeld, Germany. The mice were housed at 22 °C with a 12-h light-dark cycle (with light starting from 6:00 a.m.). Upon arrival and prior the beginning of the experiment, all mice had unrestricted access to water and a standard chow diet provided by Ssniff Spezialdieten GmbH, Soest, Germany. At the age of 12 weeks, the

mice were randomly assigned to three groups (n = 8–10, repeated in three independent experiments). The mice were then fed for a duration of 8 weeks as follows: (a) normal diet (ND) with a lipid content of 3.4% wt/wt (Rat/Mouse- Maintenance extrudate; Ssniff Spezialdieten GmbH, Soest, Germany); (b) HFD with lipid content of approximately 35% wt/wt, primarily sourced from corn oil[43]; and (c) HFD supplemented with omega-3 concentrate (HFD + F) (with 46% DHA, 14% EPA, wt/wt; product EPAX 1050 TG; Epax Norway AS, Ålesund, Norway), which replaced 15% wt/wt of dietary lipids to achieve a total EPA and DHA concentration of 30 mg/g diet[24]. The dose of omega-3 PUFAs per mouse was 0.168 g EPAX 1050 concentrate. This dose of omega-3 PUFAs in diet has been shown in the previous animal and clinical studies to prevent HFD-induced obesity complications and improve metabolic parameters[24,43,44]. The macronutrient composition of the experimental diets and fatty acid composition in dietary lipids are presented in Supplementary Tables 1, 2. Body weight and 24-h food intake were assessed weekly throughout the study period. After 8 weeks of dietary interventions, the mice were sacrificed in the fed state by cervical dislocation under diethyl ether anesthesia. Tissue samples and primary mouse BMSCs were collected for subsequent molecular analyses. All experiments were performed according to the Institute of Physiology of the Czech Academy of Sciences guidelines and were approved under protocol 81/2016.

**Glucose tolerance test**. Intraperitoneal glucose tolerance test (GTT) was performed using 1 mg of glucose/g body weight in overnight fasted mice[24]. Glucose (1 mg/g) was administered by i.p. injection. Blood glucose levels were measured from the tail vein at the indicated time (0, 15, 30, 60, 90, and 120 min), and glycemia was determined using glucometer (Accu-Chek Performa, Roche).

**Biochemical analysis**. Blood glucose levels were measured using Accu-Chek Performa glucometers from Roche. Plasma insulin levels were assessed using the Sensitive Rat Insulin RIA Kit (Millipore, Billerica, MA, USA).

**Bone histology**. Tibias were collected after the dissection and fixed for 48 h in 10% formalin. After formalin fixation, bones were demineralized by EDTA 12% solution for 14 days. Tissues were embedded in paraffin and sections were used for hematoxylin eosin staining, which was further evaluated for BMA[4].

**Isolation of BMSCs and hematopoietic stem cells (HSCs)**. BMSCs were isolated from the bones of the front and hind limbs of C57BL/6 N male mice (after 8 weeks of dietary treatments) following previous protocols with some modifications[4]. Intact bones containing BM were gently cleaned from muscles in the sterile hood. After bone-crushing, collagenase digestion (StemCell, Vancouver, BC, Canada) and negative selection of CD45, CD31 and Ter119 cells (Miltenyi, Bergisch Gladbach, Germany), BMSCs were isolated via several washing steps and centrifugation in order to get cell pellet for the culture. BMSCs were subcultured in growth medium (MEM alpha (Thermo Fisher Scientific, Waltham, MA, USA) + 20% FBS (Thermo Fisher Scientific, Waltham, MA, USA) + 1% penicillin/streptomycin (P/S) (Thermo Fisher Scientific, Waltham, MA, USA) + 0,5% Amphotericin B (Merck, Darmstadt, Germany) + 1% Glutamax (Thermo Fisher Scientific, Waltham, MA, USA) + 1% MEM NEAA (Thermo Fisher Scientific, Waltham, MA, USA) + 1% sodium pyruvate (Thermo Fisher Scientific, Waltham, MA, USA)) and cultivated for further analysis. The positive fraction with HSCs was divided into three samples. One was harvested for

mRNA isolation with Tri-Reagent (Merck, Darmstadt, Germany), the second was harvested for proteins, and the rest was seeded and cultivated in a growth medium for further analysis.

**In vitro differentiation of BMSCs**. Primary BMSCs from passage two were used for analyzing their differentiation capacity.

**Osteoblast (OB) differentiation**. BMSCs were seeded at a density of 20,000 cells/cm². When the cells reached 80% confluence in MEM medium (Thermo Fisher Scientific, Waltham, MA, USA) supplemented with 10% FBS (Thermo Fisher Scientific, Waltham, MA, USA) and 1% P/S (Thermo Fisher Scientific, Waltham, MA, USA) was added to control cells. The rest of the cells were cultivated with osteoblast induction media consisting of 10 mM β-glycerophosphate Merck, Darmstadt, Germany), 10 nM dexamethasone (Merck, Darmstadt, Germany), and 50 μg/ml Vitamin C (Wako Chemicals USA Inc., Richmond, VA, USA). The media was changed every second day for 7 days (ALP activity) and 11 days (Alizarin Red staining).

**Alizarin red staining**. Mineralization of the cell matrix at day 11 was measured using Alizarin Red S staining. Cells were fixed with 70% ice-cold ethanol for a minimum of 1 h at −20 °C, after which Alizarin Red S solution (Merck, Darmstadt, Germany) was added. Then, the cells were stained for 10 min at room temperature (RT). Excess dye was washed with distilled water, followed by PBS. The amount of mineralized matrix (bound stain) was quantified by elution of the Alizarin red stain, using 20 min of incubation of the cultures in 70% dH2O, 20% ethanol, and 10% methanol solution on a shaker (100 rpm) at RT. The absorbance of the eluted dye was measured at 500 nm using a microplate reader according to the protocol[45].

**Alkaline phosphatase (ALP) activity assay**. ALP activity and cell viability assay were quantified at day 7 of OB differentiation to normalize the ALP activity data to the number of viable cells.

Cell viability assay was performed using Cell Titer-Blue Assay Reagent (Promega, Madison, WI, USA) at fluorescence intensity ($579_{Ex}/584_{Em}$). ALP activity was determined by absorbance at 405 nm using p-nitrophenyl phosphate (Merck, Darmstadt, Germany) as substrate[46].

**Adipocyte (AD) differentiation**. Cells were plated at a density of 30,000 cells/cm². For AD differentiation, DMEM media (Thermo Fisher Scientific, Waltham, MA, USA) was used, containing 10% FBS (Thermo Fisher Scientific, Waltham, MA, USA), 9% horse serum (Merck, Darmstadt, Germany), 1% P/S (Thermo Fisher Scientific, Waltham, MA, USA), 100 nM dexamethasone (Merck, Darmstadt, Germany), 0.5 uM 3-isobutyl-1-methyxanthine (IBMX) (Merck, Darmstadt, Germany), 1 μM BRL (Merck, Darmstadt, Germany), 3 μg/mL Insulin (Merck, Darmstadt, Germany). The media was changed every three days for 10 days. Horse serum supplementation of media was used for the first three days of induction.

**Oil Red O staining**. At day 10 of differentiation, cells were rinsed with PBS and fixed in 4% paraformaldehyde (Merck, Darmstadt, Germany) for 10 min at RT. After fixation, cells were briefly rinsed with 3% isopropanol solution (Merck, Darmstadt, Germany), and lipid droplets were stained with Oil Red O solution (Merck, Darmstadt, Germany) for 1 h at RT.

**Colony forming units-fibroblast (CFU-f) assay**. After BMSC isolation, cells were seeded for CFU 500 cells/60 mm Petri dish and cultivated in growth media. After 14 days in culture, colonies

displaying more than 50 cells were counted using Crystal Violet staining (Merck, Darmstadt, Germany).

**Short-time proliferation assay**. Isolated BMSCs were plated in a 24-well plate in triplicates at a density of 1000 cells/well in a standard growth medium. Cell number was evaluated after 1, 3, 6, and 9 days. Cells were washed with PBS, detached by trypsinization, and then manually counted using a Bürker-Türk counting chamber.

**Insulin and lipopolysaccharide (LPS) responsiveness of HSCs**. Primary HSCs were cultured up to passage one and seeded for insulin and LPS stimulation. Cells were plated at a density of 300,000 cells/well in 12 well plates. When they reached the confluence, cells were starved for 4 h in serum-free MEM alpha medium (Thermo Fisher Scientific, Waltham, MA, USA) with 0.5 % BSA, and with 1% P/S then stimulated with 100 nM Insulin or 1 μg/ml LPS for 15 min at 37 °C and harvested for protein used in subsequent analyses.

**Osteoclast (OC) differentiation**. Primary mouse osteoclasts (OCs) were differentiated from total BM cells isolated from the treated mice at the end of dietary intervention[47]. Briefly, after flushing long bones and lysis of erythrocytes using erythrocyte lysis buffer, the BM cells were seeded in the cell culture plates for the subsequent analysis (96 well plates: 125,000 cell/well for TRAP staining/TRAP activity, 24 well plates: 500,000 cells/well for OC gene expression). The following day, the cells were differentiated with 25 ng/ml RANKL (Peprotech, Cat # 310-01) and 25 ng/ml M-CSF (R&D systems, cat # 216-MC) in alpha MEM media/10 % FBS/1% P/S, and the media was changed every 2nd day up to 5 days. Mature Ocs were defined as multi-nuclei cells with three or more nuclei.

**In vitro OC differentiation with short-term treatment with omega-3 PUFAs**. BM cells were isolated from C57BL/6 male mice and differentiated to OC[47]. During 5 days of OC differentiation omega-3 PUFAs: 100 μM EPA (Merck, Cat # E2011), 100 μM DHA (Merck, Cat # D2534) or their mix coupled with 0.5 % BSA were added to the differentiation cocktail and changed every 2nd day. After 5 days of OC differentiation TRAP staining and gene expression of OC markers were performed.

**TRAcP activity and TRAP staining of mature OCs**. Total BM cells were seeded in 96 well plates at the density of 125,000 cells/ well and differentiated in OC according to the protocol mentioned above up to 5 days. After day 5, media from the cultured cells was collected and used for the measurement of TRAcP activity according to the manufacture's protocol[47,48]. Then, the cells were fixed with 10% formalin and staining for TRAcP using TRAP kit (Sigma, cat. n. 387A-1KT). Mature OCs were defined as multi-nuclei cells with three or more nuclei. TRAcP activity was normalized to the number of OCs.

**Micro-computed Xray tomography (μCT) analysis**. Proximal tibias and distal body of the 5th lumbar vertebra (L5) of mice were scanned with a high-resolution μCT SkyScan 1272 (Bruker, Belgium) with resolution 3 μm per voxel (voltage 80 kV, current 125 μA with 1 mm aluminum filter, exposure 1300 ms, 2time averaging, and 0.21° rotation step on 360°scanning). Reconstruction of virtual slices was performed in NRecon 1.7.3.1 (Bruker, Belgium) with InstaRecon 2.0.4.0 reconstruction engine (InstaRecon, Urbana, IL, USA) with 49% beam hardening correction, ring artifact correction = 9, and range of intensities 0–0.09 AU for tibia and 0–0.11 for L5. Reconstructions were reoriented in DataViewer 1.5.6 (Bruker, Belgium). Areas of interest were selected based on the reference section and analyzed in CT Analyzer 1.18.4.0 (Bruker, Belgium) with structure separation based on Otsu's method. Cortical and trabecular bone were analyzed in the same area for structure volume, porosity, density, and connectivity. The analysis region was defined from the first slide under the growth plate to the 230th slide and 3D microarchitecture of trabecular and cortical bone was used for the evaluation of the bone parameters[49].

**Contrast-enhanced computed Xray tomography (CECT) workflow**
*Staining procedure*. The staining solution was prepared by mixing commercial Hexabrix® solution (Guerbet, 10 mL, 320 mgI/mL) with 1x PBS (phosphate-buffered saline, 40 mL, 10 mM)[6]. Formalin-fixed proximal tibias (right leg) of the mice were transferred to a 1.5 mL Eppendorf tube containing 1 mL of staining solution. These Eppendorf tubes were placed on a shaker plate (gentle shaking) at ambient temperature. The samples were stained for at least 3 days, after which they were scanned.

*μCT image acquisition and reconstruction*. For image acquisition, the samples were removed from the Eppendorf tube and wrapped in parafilm to prevent dehydration while exposed to X-rays. Samples were imaged using a Phoenix NanoTom M (GE Measurement and Control Solutions, Boston, MA, USA). A diamond-coated tungsten target was used. The system was operated with the following acquisition parameters: voltage = 60–70 kV, current = 120–140 μA, focal spot size = 1.99 μm, isotropic voxel size = 2 μm³, exposure time = 500 ms, frame averaging = 1, image skip = 0 and scan time = 20 min. The reconstruction was performed using Datos|x GE Measurement and Control Solutions software (version 2.7.0 – RTM) with a beam hardening correction of 8 and the inline median, ROI CT filter, and Filter volume algorithms implemented in the software. Subsequently, the datasets were normalized using an in-house developed Matlab script, with air and residual Hexabrix solution as references.

*CECT image analysis of BMAT*. After consistently aligning the datasets (DataViewer 1.5.6, Bruker MicroCT, Belgium), we analyzed the μCT data using CTAn (Bruker MicroCT, Belgium). First, we selected the volume of interest (VOI) in the proximal metaphysis of the tibia, starting 250 slices below the growth plate and covering 500 slices in the distal direction. In this VOI, binarization of the dataset was performed using a global threshold (130–255). This threshold masked both cortical and trabecular bone. Based on this selection, a denoised mask for the bone marrow combined with the trabecular bone was segmented by performing a sequence of VOI shrink-wrap, closing (2–10; increments of 2), and opening (2–10; increments of 2) operations (*i.e.*, everything inside the cortical bone was selected). Next, the newly generated mask was projected on the gray value image, generating a new VOI. In this new VOI, the segmentation of adipocytes was performed using a global threshold (1–59). For the analysis of individual adipocytes, we used the Avizo 3D (version 2021.1, Thermo Fisher Scientific, Waltham, MA, USA) software. First, the adipocytes were binarized and leftover noise was removed using interactive thresholding (1–255) and despeckle (speckle size = 7 μm × 7 μm × 7 μm) module. This was followed by segmenting individual adipocytes using a combination of thickness map computation and the H-extrema watershed module. Next, a border kill module was applied, removing adipocytes cut by the bounding box. Then, a 3D label analysis was performed that allowed the final filtering of the data based on shape (sphericity > 0.5) and volume (>4000 μm³). Finally,

sphericity, volume (µm³), area (µm²), thickness (µm), and the number of adipocytes were computed.

**Bone strength analyses**. The femora isolated from C57BL/6N male mice after 8-week-long treatment with HFD or HFD supplemented with omega-3 were tested in a three-point bending test using an ElectroForce testing system (TestBench LM1, EnduraTEC Systems Group, Bose Corp., Minnetonka, MN, USA). A standard protocol was used in this experiment[50,51]. The span length and radius of curvature of the supports were 7 mm and 2 mm, respectively. Between dissection and mechanical testing, the bones were fixed in 4% paraformaldehyde at 4 °C for the first 48 h. Then, samples were stored in PBS at 4 °C. The bones were placed with the anterior surface pointing downward and were subjected to a small stabilizing preload (1 N) and two conditioning cycles before loading until failure at a rate of 0.1 mm/s. The following parameters were derived from the load-displacement curve: (1) bone strength (N), determined as the ultimate load during the three-point bending test; (2) work-to-failure (mJ), determined as the area under the load-displacement curve, representing the energy absorbed by the bone before breaking and (3) bone stiffness (N/mm), calculated as the slope of the linear proportion of the loaded-displacement curve, representing the elastic rigidity.

**Bioenergetic analysis**. Parallel measurement of oxygen consumption rate (OCR) and extracellular acidification rate (ECAR) was performed using the Seahorse XFe24 Analyzer (Agilent, Santa Clara, CA, USA). Primary BMSCs were seeded in a 24-well Agilent Seahorse XF Cell Culture Microplate in 5 plicates at a density of 20,000 cells per well in growth media the day prior to the analysis. The next day, all wells were washed with 1 mL of DMEM (Merck, Darmstadt, Germany) supplemented with 10 mM glucose, 4 mM glutamine, and 2 mM pyruvate (pH 7.4; 37 °C); 500 µL of the same media was pipetted, and the microplate was incubated at 37 °C for 30 min. Meanwhile, an XFe24 sensor cartridge was prepared by injection of substrates according to the protocol[52] to measure metabolic rates with endogenous substrates (basal), and after subsequent additions with a final concentration of 10 mM glucose (Merck, Darmstadt, Germany), 1 µM oligomycin (Oligo) (Merck, Darmstadt, Germany), 2 µM carbonyl cyanide-4-(trifluoromethoxy) phenylhydrazone (FCCP) (Merck, Darmstadt, Germany) and mixture of inhibitors of 1 µM rotenone (Rot) (Merck, Darmstadt, Germany), 1 µg/mL of antimycin A (AA) (Merck, Darmstadt, Germany) and 100 mM 2-deoxyglucose (2DG) (Merck, Darmstadt, Germany) (2DG + AA + Rot). The Seahorse data were analyzed using Wave Software 2.6.1. (Agilent, Santa Clara, CA, USA). The data were normalized by cell number determined by Hoechst 33342 staining of cell nuclei (final concentration 5 µg/mL) (Thermo Fisher Scientific, Waltham, MA, USA), which was performed immediately after the measurement using Cytation 3 Cell Imaging Reader (BioTek, Winooski, VT, USA) and processed by Gen5 software (BioTek, Winooski, VT, USA).

**Isolation of mRNA and quantitative RT-PCR**. Total RNA exptraction was carried out using TRI Reagent (Merck, Darmstadt, Germany) and the concentration of RNA was deermined using a Nanodrop spectrometer. Subsequently, cDNA synthesis was performed from 1 µg of total RNA using a High-Capacity cDNA Reverse Transcription Kit (Thermo Fisher Scientific, Waltham, MA, USA) according to the manufacturer's protocol. Quantitative real-time PCR was conducted using Light Cycler® 480 SYBR Green I Master (Roche, Basel, Switzerland) with specific primers (GeneriBiotech, Hradec Králové, Czech Republic) as listed in Supplementary Table 3. The RT-PCR data were normalized to the expression of the housekeeping gene (36B4 for mouse).

**Western blot**. Protein lysates from the cells were prepared using M2 lysis buffer. Protein concentration was measured using BCA assay (Thermo Fisher Scientific, Waltham, MA, USA). Proteins with a final loading concentration of 15 µg/mL were separated in sodium dodecyl sulfate-polyacrylamide gels and transferred onto PVDF (polyvinylidene difluoride) membrane (Imobilon-P) by semi-dry electroblotting. After blotting, membranes were washed for 5 min in TBS (150 mM Tris-HCl, 10 mM NaCl; pH 7,4) and blocked in 5% (*w/v*) fat-free dry milk diluted in TBS-T (TBS with 1% (*v/v*) detergent Tween-20) for 1 h. After blocking, the membranes were washed 5 × 5 min in TBS-T. For immunodetection, the membranes were incubated with primary antibody (diluted in 5% milk) overnight in 4 °C. Next day, membranes were washed 5 × 5 min in TBS-T and then incubated with corresponding HRP-conjugated secondary antibody for 1 h at RT. The list of WB antibodies (Cell Signaling, Danvers, MA, USA) is presented in Supplementary Table 4. Protein detection was performed using ECL Clarity Max detection substrate (Bio-Rad) measured by ChemiDoc imaging system (Bio-Rad, Hercules, CA, USA), and signals were calculated by Image Lab software (Bio-Rad, Hercules, CA, USA). Densitometry analysis was normalized to signals from positive control protein lysates present in each membrane. Uncropped western blot membrane images are shown in Supplementary Fig. 9, 10.

**Lipidomics and metabolomics**. Global lipidomic and metabolomic profiling of BM, bone powder (BP), and plasma samples was conducted using a combined untargeted and targeted workflow for the lipidome, metabolome, and exposome analysis (LIMeX)[11,53,54] with some modifications. Extraction was performed using a biphasic solvent system of cold methanol, methyl *tert*-butyl ether (MTBE), and 10% methanol. Four different liquid chromatography-mass spectrometry (LC-MS) platforms were used for profiling: (1) lipidomics of complex lipids using reversed-phase liquid chromatography with mass spectrometry (RPLC-MS) in positive ion mode, (2) lipidomics of complex lipids in RPLC-MS in negative ion mode, (3) metabolomics of polar metabolites using hydrophilic interaction chromatography with mass spectrometry (HILIC-MS) in positive ion mode, and (4) metabolomics of polar metabolites using RPLC-MS in negative ion mode.

**Sample extraction for metabolomic and lipidomic analyses**. BM and BP samples (20–25 mg) were homogenized with 275 µL methanol containing internal standards (PE 17:0/17:0, PG 17:0/17:0, LPC 17:1, Sphingosine d17:1, Cer d18:1/17:0, SM d18:1/17:0, PC 15:0/18:1-$d_7$, cholesterol-$d_7$, TG 17:0/17:1/17:0-$d_5$, DG 12:0/12:0/0:0, DG 18:1/2:0/0:0, LPE 17:1, oleic acid-$d_9$, PI 15:0/18:1-$d_7$, MG 17:0/0:0/0:0, PS 17:0/17:0, HexCer d18:1/17:0, DG 18:1/0:0/18:1-$d_5$, TG 20:0/20:1/20:0-$d_5$, LPG 17:1, LPS 17:1, cardiolipin 16:0/16:0/16:0/16:0) and 275 µL 10% methanol containing internal standards (caffeine-$d_9$, acetylcholine-$d_4$, creatinine-$d_3$, choline-$d_9$, TMAO-$d_9$, N-methylnicotinamide-$d_4$, betaine-$d_9$, butyrobetaine-$d_9$, creatine-$d_3$, cotinine-$d_3$, glucose-$d_7$, succinic acid-$d_4$, metformin-$d_6$) for 1.5 min using a grinder (MM400, Retsch, Germany). Then, 1 mL of MTBE with internal standard (CE 22:1) was added, and the tubes were shaken for 1 min and centrifuged at 16,000 rpm for 5 min.

Plasma samples (25 µL) were mixed with 765 µL of cold methanol/MTBE mixture (165 µL + 600 µL, respectively) containing the same mixtures of internal standards as before and

shaken for 30 s. Then, 165 µL of 10% MeOH with deuterated polar metabolite internal standards was added, shaken for 30 s, and centrifuged at 16,000 rpm for 5 min.

For lipidomic profiling, 100 µL of the upper organic phase was collected, evaporated and resuspended using 100 µL methanol with internal standard (12-[[(cyclohexylamino)carbonyl]amino]-dodecanoic acid, CUDA), shaken for 30 s, centrifuged at 16,000 rpm for 5 min and used for LC-MS analysis.

For metabolomic profiling, 70 µL of the bottom aqueous phase was collected, evaporated, resuspended in 70 µL of an acetonitrile/water (4:1, $v/v$) mixture with internal standards (CUDA and Val-Tyr-Val), shaken for 30 s, centrifuged at 16,000 rpm for 5 min and analyzed using HILIC metabolomics platform. Another 70 µL aliquote of the bottom aqueous phase was mixed with 210 µL of an isopropanol/acetonitrile (1:1, $v/v$) mixture, shaken for 30 s, centrifuged at 16,000 rpm for 5 min, and the supernatant was evaporated, resuspended in 5% methanol/0.2% formic acid with internal standards (CUDA and Val-Tyr-Val), shaken for 30 s, centrifuged at 16,000 rpm for 5 min and analyzed using RPLC metabolomics platform.

**LC-MS-based lipidomics**. The LC-MS systems consisted of a Vanquish UHPLC System (Thermo Fisher Scientific, Waltham, MA, USA) coupled to a Q Exactive Plus mass spectrometer (Thermo Fisher Scientific, Waltham, MA, USA).

Lipids were separated on an Acquity UPLC BEH C18 column (50 × 2.1 mm; 1.7 µm) coupled to an Acquity UPLC BEH C18 VanGuard pre-column (5 × 2.1 mm; 1.7 µm) (Waters, Milford, MA, USA). The column was maintained at 65 °C at a flow-rate of 0.6 mL/min. For LC–ESI( + )-MS analysis, the mobile phase consisted of (A) 60:40 ($v/v$) acetonitrile:water with ammonium formate (10 mM) and formic acid (0.1%) and (B) 90:10:0.1 ($v/v/v$) isopropanol:acetonitrile:water with ammonium formate (10 mM) and formic acid (0.1%). For LC–ESI( − )-MS analysis, the composition of the solvent mixtures was the same except for the addition of ammonium acetate (10 mM) and acetic acid (0.1%) as mobile-phase modifiers. Separation was conducted under the following gradient for LC–ESI( + )-MS: 0 min 15% (B); 0–1 min 30% (B); 1–1.3 min from 30% to 48% (B); 1.3–5.5 min from 48% to 82% (B); 5.5–5.8 min from 82% to 99% (B); 5.8–6 min 99% (B); 6–6.1 min from 99% to 15% (B); 6.1–7.5 min 15% (B). For LC–ESI( − )-MS, the following gradient was used: 0 min 15% (B); 0–1 min 30% (B); 1–1.3 min from 30% to 48% (B); 1.3–4.8 min from 48% to 76% (B); 4.8–4.9 min from 76% to 99% (B); 4.9–5.3 min 99% (B); 5.3–5.4 min from 99% to 15% (B); 5.4–6.8 min 15% (B). A sample volume of 0.3 µL, 1.5 µL, and 1.5 µL was used for BM, BP, and plasma extracts, respectively, in ESI( + ). A sample volume of 5 µL for all matrices was used in ESI( − ). Sample temperature was maintained at 4 °C.

The ESI source and MS parameters were: sheath gas pressure, 60 arbitrary units; aux gas flow, 25 arbitrary units; sweep gas flow, two arbitrary units; capillary temperature, 300 °C; aux gas heater temperature, 370 °C; MS1 mass range, $m/z$ 200–1700; MS1 resolving power, 35,000 FWHM ($m/z$ 200); number of data-dependent scans per cycle, 3; MS/MS resolving power, 17,500 FWHM ($m/z$ 200). For ESI( + ), a spray voltage of 3.6 kV and normalized collision energy of 20% was used, while for ESI( − ), a spray voltage of −3.0 kV and normalized collision energy of 10, 20, and 30% were set up.

**LC-MS-based metabolomics**. Polar metabolites were separated on an Acquity UPLC BEH Amide column (50 × 2.1 mm; 1.7 µm) coupled to an Acquity UPLC BEH Amide VanGuard pre-column (5 × 2.1 mm; 1.7 µm) (Waters, Milford, MA, USA). The column was maintained at 45 °C at a flow-rate of 0.4 mL/min. The mobile

phase consisted of (A) water with ammonium formate (10 mM) and formic acid (0.125%) and (B) acetonitrile:water (95/5) with ammonium formate (10 mM) and formic acid (0.125%). Separation was conducted under the following gradient: 0 min 100% (B); 0–1 min 100% (B); 1–3.9 min from 100% to 70% (B); 3.9–5.1 min from 70% to 30% (B); 5.1–6.4 min from 30% to 100% (B); 6.4–8.0 min 100% (B). A sample volume of 0.5 µL, 0.5 µL, and 1.5 µL was used for bone marrow, bone powder, and plasma extracts, respectively, in ESI( + ). The sample temperature was maintained at 4 °C.

Polar metabolites were also separated on an Acquity UPLC HSS T3 column (50 × 2.1 mm; 1.8 µm) coupled to an Acquity UPLC HSS T3 VanGuard pre-column (5 × 2.1 mm; 1.8 µm) (Waters, Milford, MA, USA). The column was maintained at 45 °C using a ramped flow-rate. The mobile phase consisted of (A) water with formic acid (0.2%) and (B) methanol with formic acid (0.1%). Separation was conducted under the following gradient: 0 min 1% (B) 0.3 mL/min; 0–0.5 min 1% (B) 0.3 mL/min; 0.5–2 min from 1% to 60% (B) 0.3 mL/min; 2–2.3 min from 60% to 95% (B) from 0.3 mL/min to 0.5 mL/min; 2.3–3.0 min 95% (B) 0.5 mL/min; 3.0–3.1 min from 95% to 1% (B) 0.5 mL/min; 3.1–4.5 min 1% (B) 0.5 mL/min; 4.5–4.6 min 1% (B) from 0.5 mL/min to 0.3 mL/min; 4.6–5.5 min 1% (B) 0.3 mL/min. A sample volume of 5 µL was used to inject ESI( − ). The sample temperature was maintained at 4 °C.

The ESI source and MS parameters were: sheath gas pressure, 50 arbitrary units; aux gas flow, 13 arbitrary units; sweep gas flow, three arbitrary units; capillary temperature, 260 °C; aux gas heater temperature, 425 °C; MS1 mass range, $m/z$ 60–900; MS1 resolving power, 35,000 FWHM ($m/z$ 200); number of data-dependent scans per cycle, 3; MS/MS resolving power, 17,500 FWHM ($m/z$ 200). A spray voltage of 3.6 kV and −2.5 kV for ESI( + ) and ESI( − ), respectively, was used. A normalized collision energy of 20, 30, and 40% for all metabolomics platforms was used.

**Quality control**. Quality control was assured by (i) randomization of the actual samples within the sequence, (ii) injection of quality control (QC) pool samples at the beginning and the end of the sequence and between each ten actual samples, (iii) analysis of procedure blanks, (iv) serial dilution of QC sample (0, 1/16, 1/8, 1/4, 1/2, 1), (v) checking the peak shape and the intensity of spiked internal standards and the internal standard added prior to injection.

**Data processing**. LC-MS data from metabolomic and lipidomic profiling were processed through MS-DIAL v. 4.70 software. Metabolites were annotated using in-house retention time–$m/z$ library and MS/MS libraries available from commercial and open sources (NIST20, MassBank, MoNA). Lipids were annotated using LipidBlast in-built in MS-DIAL. Traces of pioglitazone ($m/z$ 355.1116) and MSDC-0602K ($m/z$ 370.0749) were detected in RPLC-MS lipidomics in ESI ( − ) as deprotonated molecules. Raw data were filtered using blank samples, serial dilution samples, and QC pool samples with relative standard deviation (RSD) < 30% and then normalized using a locally estimated scatterplot smoothing (LOESS) approach by means of QC pool samples injected regularly between ten actual samples followed by sample-weight and injection volume normalization. Data were exported as the detector signal intensity in arbitrary units (A.U.).

**Biochemical analyses of bone turnover markers**. Rat/Mouse TRAP EIA for the quantitative determination of the Tartrate-resistant acid phosphatase (TRAP) for bone resorption and Rat/Mouse P1NP EIA for the determination of the N-terminal pro-peptide of type I procollagen (P1NP) for bone formation

(MyBioSource, San Diego, CA, USA) were measured in mouse serum samples.

**Senescence-associated β-galactosidase (β-gal) activity assay**. SA-β-gal Activity 96-well assay kit (Cell Biolabs, Inc-USA, Catalog Number CBA-231) was used to measure of SA-β-Gal activity coupled with cell senescence using a fluorometric substrate according to the manufacturer protocol[46]. Analyzed cells were seeded 48 h before the assay to a black 96-well plate with a clear bottom at a density of 15,000 cells/well. All reagents were freshly prepared on the day of assay. The cells were additionally treated with 50 µM TBHP (*tert*-butyl hydrogen peroxide) for 1 h at 37 °C and 5% $CO_2$ to induce the process of senescence. Measurements were then performed in fluorescent mode (Excitation ~360 nm/Emission ~465 nm). Total protein concentration was determined by BCA protein assay kit (Merck) using absorbance mode (562 nm) in order to normalize the SA-β-Gal activity data. The results were expressed as a fold change of SA-β-gal activity.

**Cellular Reactive Oxygen Species (ROS) detection assay**. DCFDA (2,7-dichloro-dihydro-fluoroscein diacetate) (Abcam, Cambridge, United Kingdom) was used to measure the intracellular ROS production of primary BMSCs[46]. Cells were seeded to a dark, clear bottom 96-well plate at a density of 25,000 cells/well to adhere overnight. The next day, culture growth media was replaced with DCFDA Solution (25 µM) and incubated for 45 min at 37 °C and 5 % $CO_2$. DCFDA was removed and cells were loaded with 1x Buffer supplemented with 50 µM TBHP (*tert*-butyl hydrogen peroxide) for 1 h at 37 °C and 5% $CO_2$. The fluorescent intensity was detected every minute for 30 min using a fluorescent microplate reader (Excitation ~485 nm/Emission ~535 nm). The results were expressed as % of ROS production.

**Statistics and reproducibility**. Unless otherwise indicated, all data are collected from at least two independent experiments performed in triplicates. The statistical significance of the differences in the means of experimental groups was determined by unpaired t-test or ANOVA and Bonferroni or Tukey's post hoc tests using GraphPad Prism 5.0a software. The data are presented as means ± SEM. $P$ value < 0.05 was considered statistically significant.

**Reporting summary**. Further information on research design is available in the Nature Portfolio Reporting Summary linked to this article.

## Data availability

Source data for the graphs and charts are available as Supplementary Data 1 and uncropped blots are provided in Supplementary Figs. 9, 10. Any remaining information can be obtained from the corresponding author upon reasonable request.

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

## Acknowledgements

We would like to thank the Histology Core facility at CCP Biocev for their excellent work in the preparation of bone samples. We thank Kimberly Crevits at FIBEr, KU Leuven, for performing the three-point bending tests. We would like to thank Dr. Ondrej Kuda from IPHYS, CAS for his scientific input in omega-3 PUFA experiments. We are grateful for the help writing the MATLAB script for automatic histogram windowing and relative gray value normalization of CECT images to Arne Maes from KU Leuven. We are thankful for omega-3 PUFA concentrate provided by Epax Norway AS (Ålesund, Norway). The authors would like to acknowledge the Metabolomics Core Facility at the Institute of Physiology of the Czech Academy of Sciences for global metabolomic and lipidomic profiling. This work was supported by START UP Research program by the Institute of Physiology of the Czech Academy of Sciences and the Czech Science Foundation GACR 20-03586S (M.T.), GACR 19-02411S (J.K.), EFSD/NovoNordisk foundation Future leaders award (NNF20SA0066174), National Institute for Research of Metabolic and Cardiovascular Diseases (Program EXCELES, ID Project No. LX22NPO5104)—Funded by the European Union—Next Generation EU, Grant Agency of Charles University GAUK 339821 (A.B.), FRIA grant 40008717 (T.B.) and FWO G088218N (G.K. and T.B.).

## Author contributions

M.T. and J.K. conceived the project. A.B., M.F., M.T., K.B., J.F., G.A., M.D., A.C., O.H., M.R. and J.K. designed in vivo experiments, performed the in vivo and in vitro experiments, collected and analyzed data. AP and TM provided material and helped interpret bioenergetic profiling results. T.C. performed and helped with the interpretation of metabolomics data. T.B. and G.K. performed BMAT measurement in bones ex vivo and helped with BMAT evaluation, and data interpretation. WW and GHL helped to perform three-point bending tests and data interpretation. J.P. and F.S. performed μCT scanning of bones ex vivo and helped with the data analysis. M.H. helped with OC experiments. R.C. helped with animal experiment and data analysis. M.T., A.B., M.R. and J.K. designed and supervised the study and wrote the manuscript. All authors revised and approved the manuscript.

## Competing interests

All authors declare no competing interests.
