## [Peer Review File · Communications Biology]

Reviewers' comments:

Reviewer #1 (Remarks to the Author):

This well written manuscript by Benova et al provides novel insights into the effect of polyunsaturated fatty acids on bone metabolism with an emphasis on bone marrow adiposity as well as MSC and HSC function. It provides not completely unexpected, though novel findings in an animal study, supported by ex vivo experiments and detailed metabolomics. The authors have presented their results clearly and employed a wide range of technologies to prove their point but there are some concerns that should be addressed.

- 1) In the introduction, the authors refer to in vitro data that PUFAs may inhibit osteoclast function. I therefore also expected osteoclast studies in this paper. To my disappointment, the only referral to osteoclast function was indirectly. Firstly, an unconventional ratio of P1NP over TRAP in serum of the mice was reported. To me, this could reflect increased osteoblast function, decreased osteoclast function, or a combination of both. I would like to see the marker data separately, so that more can be concluded towards osteoclast function in this study. Secondly, OPG and RANKL gene expression was studied in HSCs, which also seems rather unconventional. It cannot be deduced what the expression levels were and although the expression levels may suggest a BM-localized suppression of osteoclasts, no attempts were done in this direction. The study would thus benefit from some functional work on osteoclastogenesis (TRAP staining of cultured osteoclasts, resorption, gene expression).
- 2) The introduction ends with almost an entire paragraph full of results. Apart from aim and methodologies, I would not present results here and describe these later.
- 3) The H&E images do clearly show enhanced BMAT following HFD and this also clearly reduces following the PUFA treatment, but there are clearly more BMAds in the HFD+F section than the ND, which has virtually no visible BMAds. Besides, the general cellularity seems different between these two images. Assuming the quantifications are leading for the message, I suggest to present different images (and also of more similar areas, as the epiphyses are rather different), especially for the ND group to reflect better the graphs.
- 4) Figure 4B: when looking at the wells, I do not really see the difference between the HFD and HFD+F groups. It would be good to also report on the (average) colony size.
- 5) The oil-red-O looks good but there is no info about cell numbers (for example by DAPI staining), nor is there any quantification of the oil-red-O stain. By having this information as well, statements can be made about the amount of oil-red O per cell, percentage oil-red-O positive cells but also whether cells may have undergone differential proliferation/viability during the 10 days of culture.
- 6) Both ALP and mineralization measurements would benefit from some kind of correction for cell numbers, such as total protein or DNA, again to rule out any differences in ALP or calcium deposition due to differences in cell numbers.
- 7) Figure 5: If not asked for by the journal anyway, please provide full gel images in a supplementary figure.
- 8) I would have anticipated increased senescence and ROS production following HFD, of which the first is actually confirmed by the elevated SASP gene expression. Please explain the absence of an effect on these outcomes, which are clearly counteracted by the PUFAs.
- 9) The discussion section from line 288 until 318 is written in a confusing way. It lacks coherence and appears more as a summing up of closely related studies. It is not logically built up as it first describes young animals, then old and these are then actually compared with their own study, which then is followed by yet 2 more studies. Please rewrite this paragraph in a more coherent fashion.
- 10) Line 344-345 seems to be at least partially, a repetition of the studies referred to in line 341. Please rewrite these lines to avoid overlap.
- 11) Another limitation in the discussion would be the lack of osteoclast functional data.
- 12) Line 155: refer to figure 1c-1e instead of just 1e.

Reviewer #2 (Remarks to the Author):

The manuscript submitted by Benova et al., reported the beneficial effects of Omega-3 PUFA supplementation in high fat diet (HFD+F) on bone structure and metabolism and bone marrow adipose tissue in tibia of obese mice. Moreover, bone marrow stromal cells (BMSCs) isolated from HFD+F mice display increased osteoblast and decreased adipocyte differentiation, while they also exhibit a lower energetic and senescent phenotype compared with BMSCs derived from the HFD group.

The manuscript is well written and the results are innovative and interesting. The overall structure of the manuscript is clear and logical, both for the writing and the figures. However, a few concerns arise from the experimental procedure that should be addressed accordingly.

More specifically:

At the graphs using ANOVA the a and b symbol is confusing. A comparison among all groups should be showed, in which the groups that have significant difference could be shown with a horizontal line and asterisks above. Statistical analysis should provide different symbols ie * $p < 0.05$, ** $p < 0.01$, *** $p < 0.001$. In the text, all the differences mentioned (based on a, b) are significant? If not, the significant changes should be reported.

For μ CT bone measurements of trabecular bone please also report Tb.Sp, and for cortical bone please also report cortical area fraction (B.Ar/T.Ar).

Please report measurements of bone marrow adiposity in accordance with the nomenclature and reporting guidelines from Bravenboer et al and Tratwal et al (Bravenboer et al., 2020; Tratwal et al., 2020).

The authors should explain why only males are used and what is the current state of knowledge about similar experiments in female animals.

POINT-BY-POINT RESPONSE TO REVIEWERS

COMMSBIO-23-1154

We thank the reviewers for all their thoughtful and helpful comments. We addressed all the comments raised by the reviewers. All changes are highlighted with red font in the manuscript. We believe that the enclosed revised version has improved both at the experimental levels and following extensive editing. We enclose a point-by-point rebuttal addressing the issues raised by the reviewers.

Reviewers' comments:

Reviewer #1 (Remarks to the Author):

This well written manuscript by Benova et al provides novel insights into the effect of polyunsaturated fatty acids on bone metabolism with an emphasis on bone marrow adiposity as well as MSC and HSC function. It provides not completely unexpected, though novel findings in an animal study, supported by ex vivo experiments and detailed metabolomics. The authors have presented their results clearly and employed a wide range of technologies to prove their point but there are some concerns that should be addressed.

1) In the introduction, the authors refer to in vitro data that PUFAs may inhibit osteoclast function. I therefore also expected osteoclast studies in this paper. To my disappointment, the only referral to osteoclast function was indirectly. Firstly, an unconventional ratio of P1NP over TRAP in serum of the mice was reported. To me, this could reflect increased osteoblast function, decreased osteoclast function, or a combination of both. I would like to see the marker data separately, so that more can be concluded towards osteoclast function in this study.

We thank the reviewer for this important point. We included the graphs with separate bone turnover markers in **Supplementary Fig. 2h,i**.

Secondly, OPG and RANKL gene expression was studied in HSCs, which also seems rather unconventional. It cannot be deduced what the expression levels were and although the expression levels may suggest a BM-localized suppression of osteoclasts, no attempts were

done in this direction. The study would thus benefit from some functional work on osteoclastogenesis (TRAP staining of cultured osteoclasts, resorption, gene expression).

We thank the reviewer for this important point. We performed additional *in vivo* experiment with 2-month treatment with ND, HFD and HFD+F and isolated BM cells from treated mice in order to differentiate OCs and characterize their molecular and functional properties.

The new cohort of mice after 2 months of the dietary treatment (n= 6 per group) confirmed the positive effect of omega-3 PUFAs on glucose tolerance and weight gain compared to HFD group.

We added a new paragraph in results on the effect of omega-3 PUFAs on OC differentiation (please see the lines 234-251 and Figure 5). We found that the treatment with omega-3 PUFAs showed a trend towards lower number of OCs (TRAP+ cells) compared to HFD (Fig. 5a). On the other hand, the presence of larger OCs (diameter above 500 μ m, more than 30 nuclei) was higher in HFD group, which was decreased by omega-3 PUFAs supplementation (Fig. 5b). In addition, gene expression of osteoclastic genes (*Trap* and *Ctsk*) was unchanged among groups (Fig.5e). However, gene expression of inflammatory genes such as *Il1b* (Fig.5f) was decreased by HFD+F confirming inhibitory effect of omega-3 PUFAs on OC inflammatory production.

In addition, we performed *in vitro* experiment on BM cells with acute treatment using 100 μ M EPA and DHA or the mix EPA and DHA during OC differentiation. Acute effect of EPA and DHA treatment *in vitro* during 5 days of OC differentiation confirmed an inhibitory effect of omega-3 PUFAs on OC differentiation and gene expression of OC genes (*Trap*, *Ctsk*) (please see Supplementary Fig. 6a-c)

2) The introduction ends with almost an entire paragraph full of results. Apart from aim and methodologies, I would not present results here and describe these later.

We thank the reviewer for this comment, we edited text accordingly (please see the lines 125-128).

3) The H&E images do clearly show enhanced BMAT following HFD and this also clearly reduces following the PUFA treatment, but there are clearly more BMADs in the HFD+F section than the ND, which has virtually no visible BMADs. Besides, the general cellularity seems different between these two images. Assuming the quantifications are leading for the message, I suggest to present different images (and also of more similar areas, as the epiphyses are rather different), especially for the ND group to reflect better the graphs.

We thank the reviewer for this comment. New H&E images of ND and new BMAD nomenclature parameters (Tatwal paper) are edited in new Figure 2e (please see new Figures 2e).

4) Figure 4B: when looking at the wells, I do not really see the difference between the HFD and HFD+F groups. It would be good to also report on the (average) colony size.

We thank the reviewer for the comment. The populations of the primary BMSCs used for this experiment were very heterogeneous. Therefore, we had to count individual cells in each colony and not evaluate these colonies according to the size. For example, within one well we saw colonies with smaller diameter consisting of smaller cells with a number above 50 cells per colony, and also big colonies with the larger diameter than previous type of small colony but consisting of bigger cells with total cell number under 50.

Here we are attaching the example of figures where we marked the colonies with more than 50 cells by red circle. As you can see, the number of counted colonies on these representative pictures represent the part of results from CFU-f counting in Figure 4B.

5) The oil-red-O looks good but there is no info about cell numbers (for example by DAPI staining), nor is there any quantification of the oil-red-O stain. By having this information as well, statements can be made about the amount of oil-red O per cell, percentage oil-red-O positive cells but also whether cells may have undergone differential proliferation/viability during the 10 days of culture.

We thank the reviewer for this important point. We agree that the staining with DAPI would be useful for the normalization of ORO staining. Unfortunately, the mature adipocytes were fragile after the staining and some of them detach from the plate. Thus, we were not able to perform following DAPI staining for the normalization. However, the ORO staining was confirmed by gene expression of adipogenic genes (Pparg, Cebpa, Adipoq, Fsp27 etc.) confirming lower adipogenic potential of HFD+F BMSCs vs HFD BMSCs.

6) Both ALP and mineralization measurements would benefit from some kind of correction for cell numbers, such as total protein or DNA, again to rule out any differences in ALP or calcium deposition due to differences in cell numbers.

We thank the reviewer for the comment. We normalized ALP activity to cell number measured by MTT, which is explained in graph and figure legend (please see Figure 4h).

7) Figure 5: If not asked for by the journal anyway, please provide full gel images in a supplementary figure.

We added these pictures into Supplementary Figures (please see Supplementary Fig. 7 and 8).

8) I would have anticipated increased senescence and ROS production following HFD, of which the first is actually confirmed by the elevated SASP gene expression. Please explain the absence of an effect on these outcomes, which are clearly counteracted by the PUFAs.

We thank the reviewer for this important point. As we used only 2 months of dietary treatment in C56BL/6N male mice, we did not observe a lot of changes on bone and senescent phenotype in BMSCs in HFD compared to ND condition. However, the beneficial effect of HFD+F was already pronounced in comparison to HFD condition. The animal data on obesity-induced impairment of bone phenotype are presented in 3-month HFD or longer HFD treatment in C57BL/6J male mice. We provided the explanation in Discussion part, in the limitation of the study (please see the lines 392-395).

9) The discussion section from line 288 until 318 is written in a confusing way. It lacks coherence and appears more as a summing up of closely related studies. It is not logically built up as it first describes young animals, then old and these are then actually compared with their own study, which then is followed by yet 2 more studies. Please rewrite this paragraph in a more coherent fashion.

We thank the reviewer for this comment. We edited the text and change the content according to the reviewer's suggestion (please see the lines 309-323).

10) Line 344-345 seems to be at least partially, a repetition of the studies referred to in line 341. Please rewrite these lines to avoid overlap.

We thank the reviewer for the comment. We edited the paragraph accordingly. (Please see the lines 344-348).

11) Another limitation in the discussion would be the lack of osteoclast functional data.

As we added these data in the results, we do not need to comment on it in the limitation part.

12) Line 155: refer to figure 1c-1e instead of just 1e.

We thank the reviewer for the comment. We corrected it in the Results part (please see page 5).

Reviewer #2 (Remarks to the Author):

The manuscript submitted by Benova et al., reported the beneficial effects of Omega-3 PUFA supplementation in high fat diet (HFD+F) on bone structure and metabolism and bone marrow adipose tissue in tibia of obese mice. Moreover, bone marrow stromal cells (BMSCs) isolated from HFD+F mice display increased osteoblast and decreased adipocyte differentiation, while they also exhibit a lower energetic and senescent phenotype compared with BMSCs derived from the HFD group. The manuscript is well written and the results are innovative and interesting. The overall structure of the manuscript is clear and logical, both for the writing and the figures. However, a few concerns arise from the experimental procedure that should be addressed accordingly. More specifically: At the graphs using ANOVA the a and b symbol is confusing. A comparison among all groups should be showed, in which the groups that have significant difference could be shown with a horizontal line and asterisks above. Statistical analysis should provide different symbols ie * $p < 0.05$, ** $p < 0.01$, *** $p < 0.001$. In the text, all the differences mentioned (based on a, b) are significant? If not, the significant changes should be reported.

We thank the reviewer for this important point. We added all information in the figure legends (please see pages 31-35)

For μ CT bone measurements of trabecular bone please also report Tb.Sp, and for cortical bone please also report cortical area fraction (B.Ar/T.Ar).

We thank the reviewer for this important point. We added these parameters in results in new Fig. 1f and Supplementary Fig. 2c, d, g.

Please report measurements of bone marrow adiposity in accordance with the nomenclature and reporting guidelines from Bravenboer et al and Tratwal et al (Bravenboer et al., 2020; Tratwal et al., 2020).

We thank the reviewer for this important point. We corrected the BMA parameters according to the nomenclature in the results and figures (Please see new Fig. 2b-g).

The authors should explain why only males are used and what is the current state of knowledge about similar experiments in female animals.

We thank the reviewer for this important point. We agree with the reviewer that including both sexes in the animal experiments is very important. The study was done in the collaboration with other colleagues studying AT metabolism and their design was preferable focused on male mice. For future experiments we are planning to include both males and females to perform the complete analysis studying the effect of different treatment on bone and fat metabolism.

Regarding the current state of the knowledge about similar experiments in female animals, previous studies have already documented in OVX model and aging animal model the beneficial effect of omega-3 PUFA supplementation in diet on bone phenotype and decreasing bone loss in OVX or aged female mice. However, the similar experiments in both males and females are needed for obesity-induced bone impairment and effect of dietary intervention.

Sun, D., et al. Dietary n-3 fatty acids decrease osteoclastogenesis and loss of bone mass in ovariectomized mice. *J Bone Miner Res* 18, 1206-1216 (2003).

Bhattacharya, A., Rahman, M., Sun, D. & Fernandes, G. Effect of fish oil on bone mineral density in aging C57BL/6 female mice. *J Nutr Biochem* 18, 372-379 (2007).

REVIEWERS' COMMENTS:

Reviewer #1 (Remarks to the Author):

I am satisfied with the additional experiments performed and rebuttal.

Reviewer #2 (Remarks to the Author):

The authors have addressed my comments and I propose the publication of their manuscript.